# Transmon platform for quantum computing challenged by chaotic fluctuations

Christoph Berke [1✉], Evangelos Varvelis [2,3], Simon Trebst [1], Alexander Altland [1] & David P. DiVincenzo [2,3,4]

From the perspective of many-body physics, the transmon qubit architectures currently developed for quantum computing are systems of coupled nonlinear quantum resonators. A certain amount of intentional frequency detuning ('disorder') is crucially required to protect individual qubit states against the destabilizing effects of nonlinear resonator coupling. Here we investigate the stability of this variant of a many-body localized phase for system parameters relevant to current quantum processors developed by the IBM, Delft, and Google consortia, considering the cases of natural or engineered disorder. Applying three independent diagnostics of localization theory — a Kullback–Leibler analysis of spectral statistics, statistics of many-body wave functions (inverse participation ratios), and a Walsh transform of the many-body spectrum — we find that some of these computing platforms are dangerously close to a phase of uncontrollable chaotic fluctuations.

[1] Institute for Theoretical Physics, University of Cologne, 50937 Cologne, Germany. [2] Institute for Quantum Information, RWTH Aachen University, 52056 Aachen, Germany. [3] Jülich-Aachen Research Alliance (JARA), Fundamentals of Future Information Technologies, 52425 Jülich, Germany. [4] Peter Grünberg Institute, Theoretical Nanoelectronics, Forschungszentrum Jülich, 52425 Jülich, Germany. ✉email: berke@thp.uni-koeln.de

When subject to strong external disorder, wave functions of many-body quantum systems may localize in states defined by (but not in trivial ways) the eigenstates of the disordering operators. A standard paradigm in this context is the spin-1/2 Heisenberg chain in a random $z$-axis magnetic field. Here, the disorder basis comprises the "physical" $p$-qubits defined by the spin states, different due to spin-exchange from the eigenbasis of "localized" $l$-qubits[1,2]. The latter are stationary but remain non-trivially correlated, including in the deeply localized phase.

Although it may seem paradoxical at first sight, intentional "disordering" and *many-body localization* (MBL) in the above sense are a vitally important resource in the most advanced quantum computing (QC) platform available to date, the superconducting transmon qubit array processor. Physically, the transmon array is a system of coupled nonlinear quantum oscillators. At the low energies relevant to QC the system becomes equivalent to the negative $U$ Bose Hubbard model. Site occupations 0 and 1 define the transmon $p$-qubit states, known as "bare qubits" in QC language. Randomization of the individual qubit energies maintains the integrity of these states in the presence of the finite inter-transmon coupling required for computing functionality. This coupling makes the eigen-$l$-qubits of the system different from the $p$-qubits. Considerable efforts are invested in the characterization and control of the induced correlations, known as ZZ couplings in the parlance of the QC community[3].

Connections between MBL and superconducting qubits have been considered earlier[4], but mainly with a focus on applications of qubit arrays as quantum simulators of the bosonic MBL transition. Surprisingly, however, the obvious reverse question has not been asked systematically so far: What bearings may qubit isolation by disorder have on QC functionality? Reliance on strong disorder localization is a Faustian approach inasmuch as it invites the presence of *quantum chaos*, which is an arch-enemy of quantum device control of any kind. Lowering the strength of disorder brings one closer to the MBL-to-chaos transition, heralded by the growth of $l$-qubit correlations as early indicators for the proximity of the uncontrollable chaotic phase. Since the key requirement of QC, the execution of gate operations, requires on-demand rapid growth of entanglement between $l$-qubits, it is imperative that some definite amount of coupling is present. A crucial question that we confront, therefore, is under which circumstances the necessary levels of coupling keep us outside the chaotic zone.

While this question does not have an easy overall answer, one general statement can be made with confidence: True to its Faustian nature, the invitation of disorder into the platform can be renegotiated, but not revoked. For instance, in its road map for future devices, IBM aims to replace random variations of qubit frequencies with a precision-engineered frequency alternation, e.g., … -A-B-A-B- …. While this pattern efficiently blocks resonances between neighboring qubits, *next-nearest* neighbors are now approximately degenerate. In a nonlinear system, such degeneracies are potent triggers for instabilities; the only way to control, or "localize" (qubit) states is with degeneracy lifting and translational symmetry breaking— in short, with the retention of some frequency disorder in the A and B sets.

With this general situation in mind, the purpose of this paper is twofold. In its first part, we apply state-of-the-art diagnostic tools of MBL theory to investigate the role of disorder in transmon qubit arrays. We consider realistic models of qubit arrays employed in the remarkable experimental efforts by the groups of Delft[5], Google[6], IBM[7], and others, assuming that device imperfections lead to random variations of individual qubit frequencies. Within this framework, we describe the diminishing localization

of many-$l$-qubit wave functions, and the growth of $l$-qubit correlations, upon *lowering* disorder. Considering small instances of multi-transmon systems, we find that the phase boundary between MBL and quantum chaos indeed may come dangerously close to the parameter ranges of current experiments. We also find that increasing the coordination number of the transmon lattice, as necessary for 2D connected transmon networks, increases many-body delocalization and the incipient chaos of the dynamics.

In the second part of the paper, we apply this diagnostic machinery to address the question of whether precision engineering may be employed to ultimately realize 'clean' devices. Considering the abovementioned IBM alternating sequence as a case study, we find that it may indeed be operated at low values of randomness. However, for the reasons indicated above, residual frequency variations remain required to safeguard the stability of the device; further purification will not merely lead to little further improvement, but will actually be *detrimental* to its operation. Importantly, the diagnostic framework developed in the paper may be applied to predict levels of randomness which lead to optimal localization of quantum information for given parameters characterizing the clean device.

The general conclusion of this work is that further progress towards larger QCs will be dependent on skirting the dangerous attributes of chaotic parts of the parameter space. We know from experience with general many-body systems that tendencies to long-range correlations and delocalization increase with increasing two-dimensional system connectivity[8]. On this basis, the monitors provided by the many-body localization theory may become an essential resource in the perfection of future transmon-based information devices.

## Results

**Overview.** In what follows, we introduce our principal object of study: a transmon array modeled with realistic qubit parameters. Anticipating the importance of nonlinearities, we use effective "low-energy Hamiltonians" solely to gain an intuition of the underlying physics but perform all subsequent computations avoiding any such approximations. We then introduce diagnostics inspired by MBL theory and apply them to detect signatures of chaos. We discuss enhanced tendencies to instability emerging in two-dimensional geometries and address the question of whether the ideal of a stable and "perfectly clean" array can be reached by advanced qubit engineering.

**Transmon array Hamiltonian.** Our study begins with the well-established minimal model for interacting transmon qubits[9,10]:

$$H = 4E_C \sum_i n_i^2 - \sum_i E_{J_i} \cos \phi_i + T \sum_{\langle i,j \rangle} n_i n_j. \quad (1)$$

Here, $n_i$ is the Cooper-pair number operator of transmon $i$, conjugate to its superconducting phase $\phi_i$. The transmon charging energy $E_C$ is determined by the capacitance of the metal body of the transmon and is easily fixed at a desired constant, typically about $E_C = 250$ MHz ($h = 1$). The Josephson energy $E_J$ is proportional to the critical current of the junction. Except in very recent work, it has been difficult to fix this constant reproducibly to better than a few percent. However, typical values lie in the vicinity of around 12.5 GHz, much larger than the charging energy. Finally, electrical coupling between the transmons, often via a capacitance, produces the charge coupling $T n_i n_j$. The coefficient $T$ has varied over a substantial range in 15 years of experiments[11]; $T$ values beyond 50 MHz are possible, but $T < E_C$ is a fundamental constraint. Current experiments are often in the range $T = 3$–5 MHz, making $T$ the smallest energy scale in the problem.

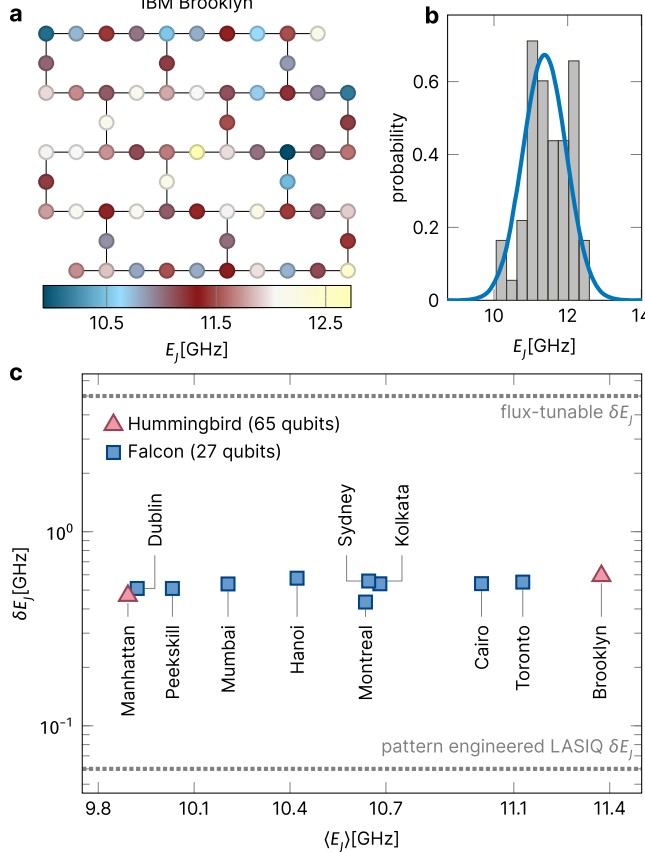

**Fig. 1 Experimental parameters of recent IBM transmon arrays. a** Layout of the 65-qubit transmon array "Brooklyn", currently available in IBM's quantum cloud (https://www.ibm.com/quantum-computing/systems/), in a heavy-hexagon geometry. The coloring of the qubits indicates the variation of Josephson energies $E_J$ which is largely uncorrelated in space. **b** Spread of the $E_J$ plotted for the "Brooklyn" chip, consistent with a Gaussian distribution (solid line). Similar levels of disorder and distributions are found in all transmon devices available in IBM's quantum cloud. **c** Variance of the measured Josephson energies, $\delta E_J$, for nine realizations of the 27-qubit "Falcon" design, and two realizations of the 65-qubit "Hummingbird" design. While the mean varies unsystematically from device to device, the variance remains very consistent, setting the parameter favored in our "scheme A" study below. "Scheme B" cases in other labs have a much larger spread as indicated by the "flux tunable" level in the figure. Recent proposals of using high precision laser-annealing[12] as a pattern engineering approach[13], discussed towards the end of the manuscript, aim for a significant reduction of the $E_J$ variance; such pattern-tuned transmon arrays have so far not appeared in any cloud device.

This model has been very concretely realized in experiments in many labs in recent years, but notably also in the many chips that have been made available for use in the IBM cloud service (https://www.ibm.com/quantum-computing/systems/). These devices of the "Falcon" and "Hummingbird" generations have employed transmons laid out in the heavy-hexagon lattice geometry of Fig. 1a. While these devices have fixed values of the coupling parameter $T$ and of the charging energy $E_C$, their Josephson energy $E_J$ varies from transmon to transmon. This effectively random variation is, in fact, crucially required to prevent the buildup of inter-transmon resonances, and the compromising of quantum information; its role in the physics of present-day transmon device structures, with insights drawn from many-body localization theory, is the central theme of this paper.

Before addressing the physics of the full model Eq. (1), let us consider its low-energy limit. Applying a sequence of approximations (series expansion of the Josephson characteristic, rotating wave approximation) one arrives at the effective Hamiltonian

$$
\begin{aligned}
H = \sum_i \nu_i a_i^\dagger a_i &- \frac{E_C}{2} \sum_i a_i^\dagger a_i (a_i^\dagger a_i + 1) \\
&+ \sum_{\langle i,j \rangle} t_{ij}(a_i a_j^\dagger + a_i^\dagger a_j),
\end{aligned}
$$
$$
\nu_i \equiv \sqrt{8 E_{J_i} E_C}, \qquad t_{ij} = \frac{T}{4\sqrt{2 E_C}} \sqrt[4]{E_{J_i} E_{J_j}}.
$$

(2)

To leading order, this model describes the transmon as a harmonic oscillator, where the above choices of energy scales place the frequencies $\bar{\nu}_i \approx 6$ GHz on average in the middle of a microwave frequency band, convenient for precision control. The attraction term, a remnant of the cos-nonlinearity, is considerably smaller than the average harmonic term, which is desired for transmon operation. Finally, the characteristic strength of the nearest neighbor hopping coefficients, $|t_{ij}| \approx \frac{T}{4\sqrt{2}}\sqrt{\frac{E_J}{E_C}}$ (often called $J$ in the literature), continues to be the smallest energy scale in the problem.

Unless novel engineering techniques are used (see below), the abovementioned variations of the Josephson energy, $\delta E_J$, are in the few percent range; thus, at a minimum, there is variation in oscillator frequencies $\nu_i$ of around $\delta \nu_i \approx (\bar{\nu}/2 E_J) \delta E_J \approx 120$ MHz, when the typical $E_J \approx 12.5$ GHz. This scale is much larger than the particle hopping strength, which for the same parameter set is about $|t_{ij}| \approx 6$ MHz.

From the perspective of many-body physics, these variations make Eq. (2) a reference model for bosonic MBL. For the above characteristic ratio $\delta \nu / |t| \sim 20$ we can hope that the system is in the MBL phase, and we confirm this below. From the perspective of transmon engineering, the frequency spread blocks the buildup of local nearest neighbor or next-nearest neighbor inter-qubit resonances. Below, we will discuss how these two perspectives go together (and where they may depart from each other). However, before turning to the observable consequences of frequency spread, we note that there exist two broad design philosophies for its realization in transmon array structures, schemes A and B throughout.

Generally speaking, scheme A aims to suppress the frequency spread to the lowest possible values required for the stability of the structure, or dictated by limits in precision engineering. For example, Fig. 1b shows that the spread of Josephson energies in IBM devices is consistent with a Gaussian distribution (with no stringent correlations from site to site). These observations hold true for all current devices whose parameters are documented publicly by IBM. Figure 1c shows that the variance $\delta E_J$ has in fact remained very constant over 9 realizations of "Falcon" chips (27 qubits) and the two latest "Hummingbird" chips (65 qubits). This 'natural disorder' regime was in use in many generations of quantum computer processors[7] that IBM has provided on its cloud service since 2016. However, a significant reduction of disorder has been reported in a very recent line of research at IBM employing high precision laser-annealing[12] as a pattern engineering approach[13] to be discussed towards the end of the manuscript.

The complementary scheme B embraces frequency disorder as a potent means of protecting qubit information, and in fact, works to effectively *enhance* it. Examples in this category include the recent reports from TU Delft on their extensible module for surface-code implementation[5]. Google devices such as its 53-qubit processor[6] contain engineered frequency patterns whose aperiodic variation effectively realizes a form of the synthetic

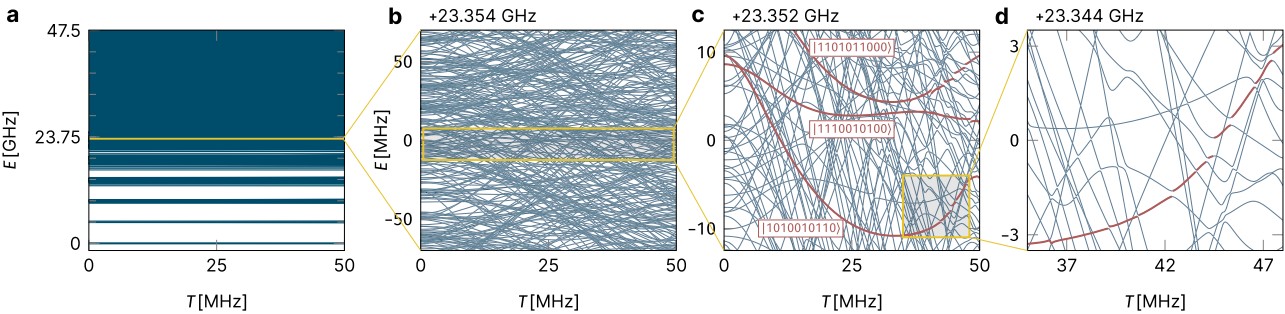

**Fig. 2 Energy spectrum of a coupled transmon array.** Illustrated are the energy levels $E_\alpha(T)$ of Hamiltonian Eq. (1) on varying energy scales; the data shown is for a coupled transmon chain of length $N = 10$. Panel **a** illustrates the clustering of levels into energy bundles corresponding to the total number of bosonic excitations. Panel **b** zooms into the five-excitation band, which upon further enlargement in panel **c** reveals level repulsions that become particularly visible for larger couplings. In panel **c** we also mark in red a number of computational states (identified in this energy window at vanishing coupling $T = 0$). The further zoom-in of panel **d** traces one such computational state through a sequence of avoided level crossings.

disorder, where in addition the qubit coupling $t_{ij}$ is lowered during idle periods. In this way, the ratio $\delta v/t$—the relevant scale for localization properties—is drastically enhanced.

Below, we will consider both schemes A and B, and investigate the incipient quantum localization and quantum chaos present in the two settings. In our model calculations, we represent the disorder by drawing independent samples from a Gaussian distribution with standard deviation $\delta E_J$, added to the mean Josephson energy $E_J$. As a representative A case we take $\delta E_J \approx 500$ MHz (for a Josephson energy of $E_J = 12.5$ GHz, giving $\delta v \approx 120$ MHz, as above), while for a B case we will take $\delta E_J$ and $\delta v$ some ten times larger (precise numbers are given below, see also Supplementary Note 1 for further discussion of experimental parameters). Note that from this point onward, we continue with the full model Eq. (1).

Specifically, for scheme-A parameter ranges, the energy eigenvalues of Eq. (1) cluster into energy bundles corresponding to the total number of bosonic excitations, as seen in Fig. 2a. Looking inside the 5-excitation band, we see (in Fig. 2b) a dense tangle of energy levels. However, only some of these levels are used to perform quantum computations in quantum processors; the identification of these levels, as shown in Fig. 2c and discussed in detail below, can only be done unambiguously if we are far away from the chaotic phase.

Having QC applications in mind, we are primarily interested in signatures of quantum chaos in the "computational subspace" of the bosonic Hilbert space, i.e., the space comprising local occupation numbers $a_i^\dagger a_i = 0, 1$, defining the $p$-qubit states for QC. In that Hilbert space sector, the problem reduces to a disordered spin-1/2 chain, another paradigm of MBL. Recent results from the MBL community indicate that the separation into a chaotic ergodic and an integrable localized phase is not as straightforward as previously thought, and that wave functions show remnants of extendedness and fractality even in the 'localized' phase[14].

**Diagnostics.** In the following, we analyze the Hamiltonian Eq. (1) with a combination of different numerical methods tailored to the description of localized phases:

- *Spectral statistics:* According to standard wisdom, many-body spectra have Wigner–Dyson statistics in the phase of strongly correlated chaotic states, and Poisson statistics in that of uncorrelated localized states[15]. Real systems show more varied behavior, quantified below in terms of a Kullback–Leibler divergence (see Methods for details). This produces a charting of parameter space indicating the

chaos/MBL boundary and the rapidity with which the boundary is approached.
- *Wave function statistics:* Focusing on the localization regime, we analyze how strongly the eigenstates differ from the localized states of the strictly decoupled system.
- *Walsh transforms:* We quantify the correlations between $l$-qubits (known in the QC community as ZZ couplings, and in the MBL community as $\tau$-Hamiltonian tensor coefficients) by application of a Walsh transform filter. To the best of our knowledge, this particularly sensitive tool has not been applied so far to the diagnostics of MBL.

We consider a system of $N$ coupled transmons in a one-dimensional chain geometry—a minimalistic setting that allows us to probe essential aspects of localization physics and quantum chaos using the above diagnostics and whose computational feasibility allows us to map out the broader vicinity of experimentally relevant parameter regimes. Typical system sizes vary between $N = 5$ and 10 sites, as detailed below.

**Spectral statistics.** We probe the spectral signatures of this coupled transmon system in an energy bundle of excited states (see Fig. 2b), which are generated by a total of $N/2 = 5$ excitations. For the $N = 10$ transmon chain at hand, this manifold contains a total of 2002 different states. States within this bundle that have local excitation numbers 0,1 can be viewed as typical representatives in the computational subspace. Zooming in on this mid-energy spectrum, we plot its spectral statistics in the main panel of Fig. 3: The KL divergence vanishes when calculated with respect to the Poisson distribution for small transmon couplings, indicating perfect agreement with what is expected for an MBL phase. This is also corroborated by the striking visual match of the distributions in the corresponding inset of Fig. 3. But the KL divergence is maximal when compared to Wigner–Dyson statistics (red curve in Fig. 3). This picture is inverted for large transmon couplings $T \approx 70$ MHz, where we find an extremely good match to Wigner–Dyson statistics—unambiguous evidence for the emergence of strongly correlated chaotic states. Probably even more important is the fact that these KL divergences allow us to quantify proximity to the diametrically opposite regimes for all intermediate coupling parameters. This includes a region of 'hybrid statistics' around the crossing point of the two curves, indicating an equal distance from both limiting cases, which we will discuss in more detail below.

By way of this KL divergence one can then map out an entire phase diagram, e.g., in the plane spanned by varying values of the transmon coupling and Josephson energy, while fixing the

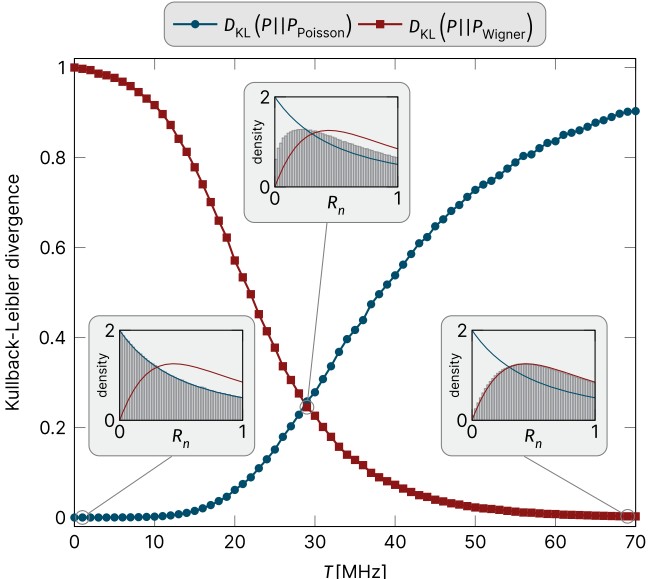

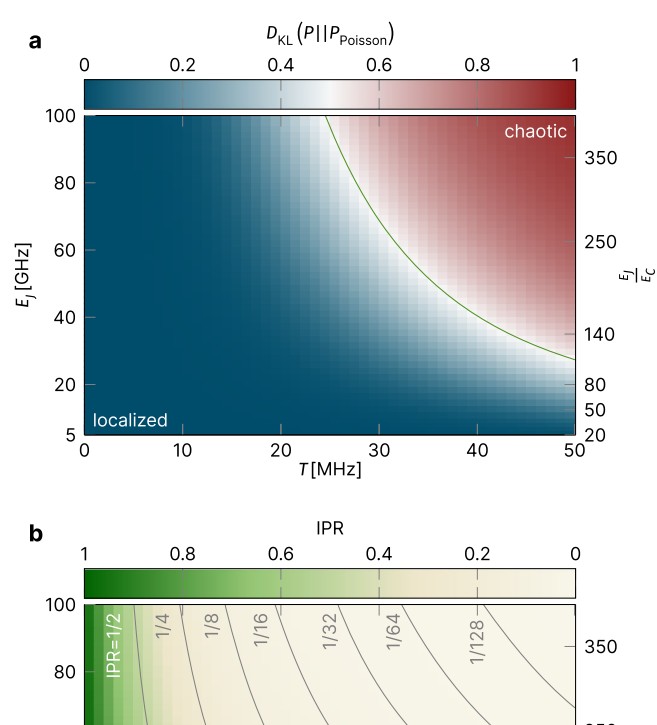

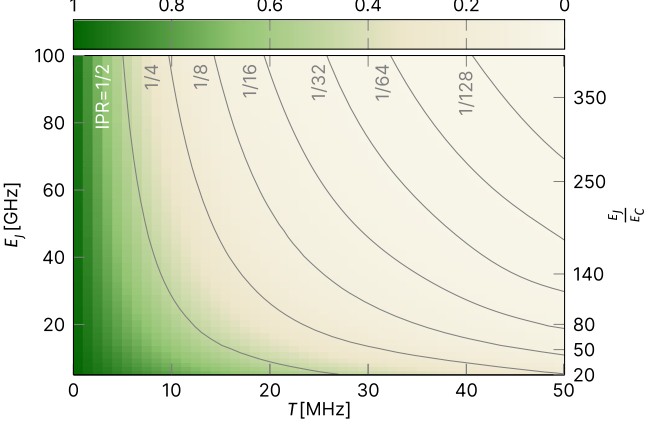

**Fig. 3 Spectral statistics.** Shown are data for a chain of $N = 10$ transmons versus the coupling parameter $T$. The average Josephson energy is fixed to $E_J = 44$ GHz, and scheme-A disorder is used ($\delta E_J = 1.17$ GHz). These statistics indicate a transition from Poisson statistics (blue) in the MBL regime (at low coupling) to Wigner–Dyson statistics (red) in a many-body delocalized regime (at large coupling). Shown are normalized Kullback–Leibler (KL) divergences $D_{KL}$ (see Eq. (7) in Methods) calculated for the distribution of ratios of consecutive level spacings $R_n$ in the energy spectrum, such as the ones illustrated in the insets for three characteristic couplings. The KL divergences are normalized such that $D_{KL}(P_{Wigner} \| P_{Poisson}) = 1$ and vice versa. All results are averaged over at least 2500 disorder realizations.

charging energy as shown in Fig. 4 (for scheme-A parameters). This allows us to clearly distinguish the existence of two regimes, the expected MBL phase (colored in blue) for small transmon coupling and a quantum-chaotic regime (colored in red), where the level statistics follow Wigner–Dyson behavior (with delocalized, but strongly correlated states) for sufficiently strong transmon couplings. It is this latter regime that one surely wants to avoid in any experimental QC setting. But before we discuss the experimental relevance of our results, we want to characterize more deeply the quantum states away from this chaotic regime using additional diagnostics.

**Wave function statistics.** One particularly potent measure of the degree to which a given wave function is localized or delocalized, is its inverse participation ratio (IPR), i.e., the second moment

$$\text{IPR} = \sum_{\{n\}} \langle |\psi_n|^4 \rangle, \qquad (3)$$

where the angular brackets denote averaging over disorder realization, and $\sum_{\{n\}}$ is a symbolic notation for the summation over a chosen basis (in the present case, the Fock basis). An IPR of 1 indicates a completely localized state (as in our example for vanishing coupling $T = 0$), while an IPR less than 1 indicates the tendency of a wave function towards delocalization[16].

Here we consider the IPR measured as an average over all states in one of the energy bundles illustrated in Fig. 2a, e.g., the manifold of typical states with $N/2 = 5$ bit flips considered in the spectral statistics above. Figure 4b shows the IPR in the same parameter space as in (a). What is most striking here is that the IPR rapidly decays—the contour lines in the panel indicate exponentially decaying levels of 1/2, 1/4, 1/8, …1/128—showing

**Fig. 4 Phase diagrams. a** summarizes the *spectral statistics* by plotting the Kullback–Leibler divergence $D_{KL}$ with respect to the Poisson distribution in the plane spanned by the Josephson energy $E_J$ and the transmon coupling $T$ for scheme-A parameters. One identifies an MBL regime (blue) for small couplings and a quantum-chaotic regime (red) following Wigner–Dyson statistics for large couplings. **b** summarizes the *wave function statistics* for the same parameters by color-coding the inverse participation ratio (IPR) showing a fast drop to values below 1/2 already for moderate coupling strength. The gray lines indicate contour lines of constant IPR. All results are averaged over at least 2000 disorder realizations. The spread of the Josephson energies varies from $\delta E_J \sim 0.4$ GHz for $E_J = 5$ GHz to $\delta E_J \sim 1.7$ GHz for $E_J = 100$ GHz (see Methods for details).

that the wave functions quickly delocalize. Note in particular, that the IPR has dropped to a value of less than 10% in the region of "hybrid statistics" identified in the level spectroscopy above.

**Walsh-transform analysis.** The MBL phase is the right place to be for quantum computing since computational qubits (the *l*-qubits above) retain their identity there. But, as indicated by the drop in IPR, even the localized phase may be problematic. We, therefore, apply another diagnostic that is specifically adapted to identifying problems with running a quantum computation in the MBL phase. It begins with the expectation, announced in[1,2], that the Hamiltonian of the multi-qubit system, in the *l*-qubit basis, can be expressed as

$$H = \sum_i h_i \tau_i^z + \sum_{ij} J_{ij} \tau_i^z \tau_j^z + \sum_{i,j,k} K_{ijk} \tau_i^z \tau_j^z \tau_k^z + \dots \qquad (4)$$

$$= \sum_{\mathbf{b}} c_{\mathbf{b}} Z_1^{b_1} Z_2^{b_2} \dots Z_N^{b_N}. \qquad (5)$$

Eq. (4), the "$\tau$-Hamiltonian" of MBL theory[1,2], embodies the observation that a diagonalized Hamiltonian can be written in a basis of diagonal operators $\tau_i^z$, which are the same as the Pauli-Z operators ($Z_i$) in the quantum-information terminology of Eq. (5). Here the sum is over $N$-bitstrings $\mathbf{b} = b_1 b_2 \ldots b_N$, where each $b_i$ is 0 or 1.

A system described by the $\tau$-Hamiltonian can be an excellent data carrier for a quantum computer, particularly if the high-weight terms are small. If only the one-body terms in Eq. (4) are non-zero, the system is an ideal quantum memory: In the interaction frame, defined by the non-entangling unitary transformation $U(t) = \exp(it \sum_i h_i \tau_i^z)$, all quantum states, including entangled ones, remain stationary. Unfortunately, the expectation of MBL theory is that the two-body and higher interaction terms are non-zero and grow as the chaotic phase is approached.

We have performed a numerical extraction of the parameters of Eqs. (4–5) for a five-transmon chain. We find that problematic departures from full localization do indeed occur already at rather small values of the qubit-qubit coupling parameter $T$. This reinforces the message, in a *basis-independent* way, of our IPR study. But this extraction must begin with a very non-trivial step, namely the identification of the qubit eigenenergies of the transmon Hamiltonian. Since this Hamiltonian Eq. (1) is bosonic, it has a much larger Hilbert space than the spin-1/2 view embodied in Eqs. (4)–(5). The qubit states, those with bosonic occupations limited to 0 and 1, are not separated in energy from the others but are fully intermingled with states of higher occupancy, as illustrated in Fig. 2c. It would thus appear that this truncation is rather unnatural—but it is in fact crucial to the whole quantum computing program with transmons. It is essential to pick out, from all the eigenlevels $E_\alpha$ of the full Hamiltonian Eq. (1) as shown in Fig. 2, just the subset of levels $E_\mathbf{b}$ that can be associated with a bitstring label $\mathbf{b}$ (cf. Eq. (5)).

Having performed such a state identification (as discussed in the Methods) and tagged the subset of eigenlevels $E_\mathbf{b}(T)$ that can be identified as qubit states, the coefficients of the $\tau$-Hamiltonian are easily obtained by a Walsh–Hadamard transform[17]:

$$c_\mathbf{b}(T) = \frac{1}{2^N} \sum_{\mathbf{b}'} (-1)^{b_1 b_1'} (-1)^{b_2 b_2'} \ldots (-1)^{b_N b_N'} E_{\mathbf{b}'}(T)$$
$$= \frac{1}{2^N} \sum_{\mathbf{b}'} (-1)^{\mathbf{b} \cdot \mathbf{b}'} E_{\mathbf{b}'}(T). \quad (6)$$

Being a kind of Fourier transform, the Walsh transform functions to extract correlations, in this case in the correlations of the computational eigenstates (in energy). Figure 5 shows these coefficients vs. $T$. For small $T$, many of the expectations from MBL theory[1,2] are fulfilled: There is a clear hierarchy according to the locality of the coefficients. Thus, nearest-neighbor ZZ interactions are the largest, followed by second-neighbor ZZ and contiguous ZZZ couplings, and so forth. Jumps occur in these coefficients, initially very small, which arise from the switching of labeling at anticrossings.

The two-body ($J_{ij}$/ZZ) terms are known and carefully analyzed in transmon research[3,18,19]. Their troublesome consequences, including dephasing of general qubit states, and failure to commute with quantum gate operations, add overhead which is ultimately found to be insupportable, enforcing a practical upper limit of $J_{ij} \sim 50–100$ kHz. This limit, marked (dashed line) in Fig. 5b, is exceeded already at $T = 3$ MHz. Even though the transition to chaos is still a long way off, quantum computing becomes very difficult above this limit.

**Scheme B: spread frequency distribution.** Other techniques for executing entangling gates leave considerably more freedom to

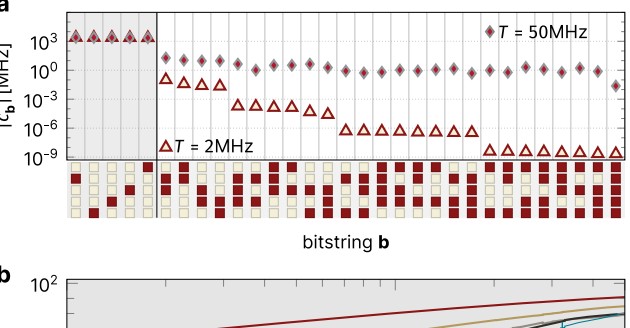

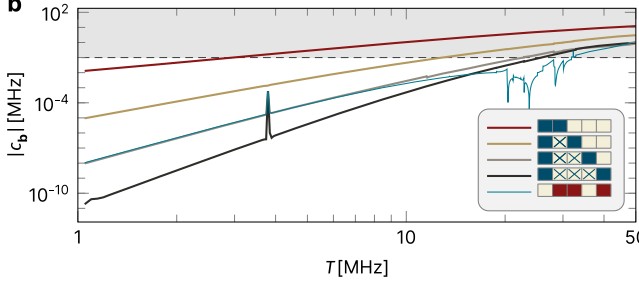

**Fig. 5 Walsh-transform analysis. a** Comparison of the $c_\mathbf{b}$ coefficients of Eq. (6) for a five-qubit chain with scheme-A parameters, for two values of the coupling $T$. Along the $x$-axis, are the 31 different values of the bitstring $\mathbf{b}$ with at least one non-zero bit. We use a graphical depiction of each bitstring, as a vertical column of five boxes, so that the first bitstrings, starting from the left, are 01000, 00001, 00010, etc. With this graphical depiction, one can see immediately which of the five transmons are involved in the given $\tau$-Hamiltonian coefficient. The $|c_\mathbf{b}|$ are sorted from largest to smallest for the $T = 2$ MHz data, which reveals a clear hierarchy of strengths according to the maximal distance between two 1's in the bitstring. There is no such systematic behavior for the large-$T$ case (plotted for the same ordering). **b** Absolute value of averaged Walsh coefficients as a function of the coupling $T$. The inset introduces a new coloring notation for the bitstrings $\mathbf{b}$; blue-colored bit boxes indicate that the $c_\mathbf{b}$ shown is averaged over all cases with the same maximal distance of two 1's. The convention is explained more fully in the Methods section, Fig. 9. Also shown for comparison is the absolute value of the Walsh coefficient for the specific bitstring $\mathbf{b} = 01101$. The dashed line and the shading above mark the "danger zone" $|c_\mathbf{b}| \gtrsim 100$ kHz indicated by recent experimental studies on ZZ coupling[3].

increase the disorder, with $\delta E_J$ values in the GHz range. This option can forestall the growth of problematic precursors of chaotic behavior. A good example of a quantum computer that uses this freedom is the surface-7 device of TU Delft[5]. During gate operation, the qubit frequencies are temporarily tuned into resonant conditions that "turn on" entanglement generation. This is done in a pattern that does not lead to any extensive delocalization. In the 53-qubit quantum computer of Google[6], this tuning is also available, but in its operation, an additional strategy is used: extra hardware is introduced to also make $T$ tunable. Being able to set the effective $T$ to zero (although only in a perturbative sense) of course eliminates the problem of delocalization, and in this latest Google work, $\delta E_J$ has been returned to a small value. Google made major changes in its "Hamiltonian strategy" in recent years[6] that have led them to their recent success. In Supplementary Note 2, we discuss scheme-B parameters (as found in recent Delft chips[5]) quantitatively using our three diagnostics.

**Transmon arrays in higher dimensions.** All the conclusions of the last few sections have been reached by calculations for transmons coupled in a one-dimensional chain geometry. However, actual quantum information architectures are two dimensional, and we have therefore also simulated the surface-7

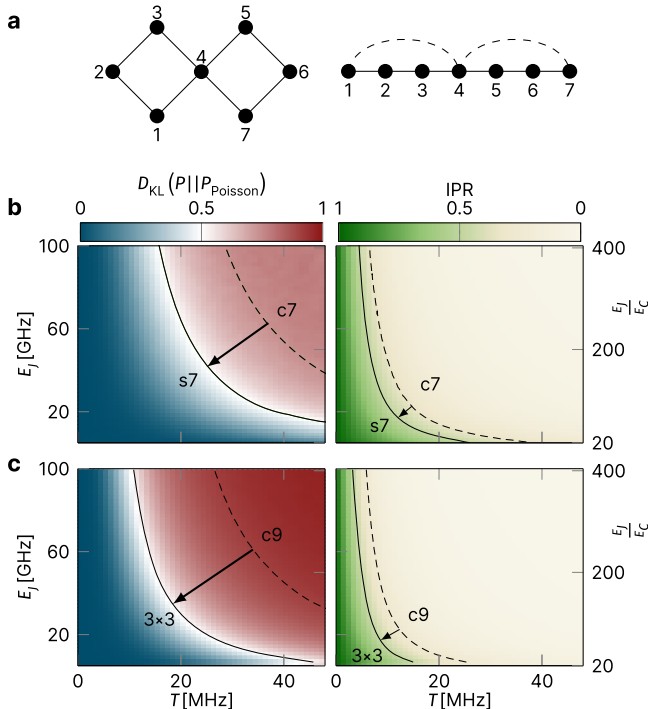

**Fig. 6 Two-dimensional transmon geometries. a** Surface-7 (s7) geometry. **b** Phase diagram of seven coupled transmons, coupled in a surface-7 geometry. The inclusion of two additional couplings in comparison to a chain of seven transmons (c7) leads to significant shifts in the phase diagram calculated for scheme-A parameters, as illustrated in the left panel for the level statistics and in the right panel for the IPR. Shown on the left is the shift of the line indicating where the normalized KL divergence with regard to Poisson statistics $D_{KL}$ has increased to 0.5 (see also Figs. 3 and 4). On the right, we indicate the shift of the line indicating where the IPR drops below 0.5, akin to the lower panel in Fig. 4. All results are averaged over at least 1500 disorder realizations. **c** Phase diagram of a $3 \times 3$ transmon array. All results are averaged over at least 2500 disorder realizations. For both geometries, the same scheme-A parameters as in Fig. 4 were used.

layout[5,20] as well as a $3 \times 3$ transmon array as minimal examples in this category. The surface-7 chip comprises a pair of square plaquettes, which is obtained from a chain of seven transmons by including two additional couplings, see top panel of Fig. 6. (Google's 53-qubit layout extends this principle to a large square array extension.)

The study of two-dimensional geometries, or of one-dimensional arrays shunted by long-range connectors, is motivated by the realization that the case of strictly one-dimensional chains is exceptional: in one dimension, "rare fluctuations" with anomalously strong local disorder amplitudes may block the correlation between different parts of the system, enhancing the tendency to form a many-body localized state. In higher dimensions, such roadblocks become circumventable, which makes disorder far less efficient in inhibiting quantum transport. For an in-depth discussion of the effects of dimensionality on MBL, we refer to ref. [8].

Our simulations of the surface-7 and $3 \times 3$ architectures, where we have chosen scheme-A parameters, are summarized in Fig. 6b, c. The spectral and wave function statistics data indicate that, not surprisingly, chaotic traces are rather more prominent than in the one-dimensional simulations (and despite the fact that to get the surface-7 geometry, we nominally added only two extra couplings in comparison to a purely one-dimensional geometry, see Fig. 6a). The bottom line is that the comfort zone introduced by

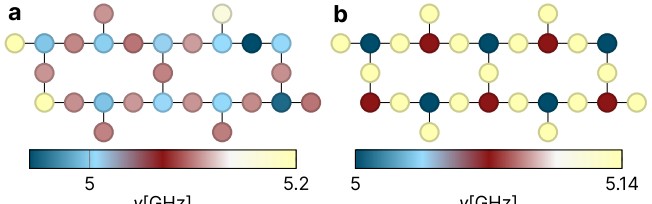

**Fig. 7 Frequency alternation patterns. a** A-B-A-B pattern. **b** A-B-A-C pattern optimized for scheme-A architectures on the heavy-hexagon geometry. Note that while **b** is an idealized structure[12,27], the A-B pattern of **a** is one that is currently implemented in experiment[13], and thus with a slightly imperfect setting of frequencies, visible especially on the far left and right of the device.

disorder schemes is considerably diminished when including higher-dimensional couplings.

**Qubit frequency engineering**. Until very recently, process variations in $E_J$ have led to an inevitable spread in qubit frequencies, as described by the effectively Gaussian distributions employed above (see Fig. 1b). However, the development of a high precision laser-annealing technique[12] (LASIQ, see Fig. 1c and Supplementary Note 3) has changed the situation and is opening the prospect to clone qubits with unprecedented precision. IBM proposes[13] to use this freedom and realize arrays with A-B-A-B, or A-B-A-C (Fig. 7) frequency alternation, effectively blocking unwanted hybridization between neighboring qubits. However, even then a residual amount of random frequency variation remains essential for the functioning of the device. For example, a perfect A-B-A-B sequence would block nearest neighbor hybridization at the expense of creating dangerous resonance between the *next-nearest* qubits; more formally, perfectly translationally invariant arrays would have extended Bloch eigenstates, different from the localized states required for computing.

The question thus presents itself as to how to optimally navigate a landscape defined by the extremes of Bloch extended, chaotic, and many-body localized wave functions for absent, intermediate, and strong disorder, respectively. In Supplementary Note 4, we apply the diagnostic framework introduced earlier in the paper to address this question in quantitative detail. To summarize the results, we observe that for diminishing disorder the transmon-array Fock space disintegrates into a complex system of mutually decoupled subspaces, reflecting the complexity of the A-B or A-B-A-C "unit cells". The strength of the residual disorder determines whether wave functions are chaotically extended or localized within these structures. Employing the inverse participation ratio as a quality indicator, we find that recent IBM engineering succeeded in hitting the optimum of near localized states with IPRs close to unity. However, it is equally evident that further reduction of the disorder would delocalize these states over a large number of qubit states and be detrimental to computing; disorder remains an essential resource, including in devices of the highest precision.

## Discussion
The subject of this study has been an application of the state-of-the-art methodology of many-body localization theory to realistic models of present-day transmon computing platforms. In the mindset of the localization theorist, all transmon quantum information hardware has in common that it operates in a regime where the tendency of quantum states to spread by inter-qubit coupling is blocked by the detuning of qubit frequencies—the many-body localized phase. Within this phase, there is considerable freedom for the realization of localization-protected

architectures; different strategies include A: weak coupling at weak detuning (IBM), or B: strong, intentionally introduced detuning interspersed by sporadic and incomplete coupling (Delft/Google). In this background, we have explored the integrity of different localized phases, in view of the omnipresent phase boundary to a chaotic sea of uncontrollable state fluctuations in the limit of too weak detuning and/or too strong coupling.

The single most important insight of this study is just how extended the twilight zone of partially compromised quantum states already is before reaching the boundary to hard quantum chaos. One may object that the existence of a crossover zone is owed to the smallness of the transmon arrays of 5–10 units studied in this work. However, it has to be kept in mind that the dimension of the random Hilbert spaces in which the many-body quantum states live is exponential in these numbers and large by any (numerical) standard of localization theory. This indicates that 'finite-size effects' in these systems are notorious and must be kept in mind for computing architectures of technologically relevant scales.

A second unexpected finding is that early indicators of chaotic fluctuations show in different ways in different observables. Among these, the least responsive observable is many-body spectral statistics, the most frequently applied diagnostics of the MBL/chaos transition. However, the computational states themselves respond far more sensitively to departures from the limit of extreme localization. We have observed this tendency in the standard observable for wave function statistics, the inverse participation ratios, where Fig. 4 shows tendencies to strong wave function spreading already in parameter regimes where spectral statistics suggests complete safety. Surprisingly, however, the Walsh transform diagnostic—which is uncommon in MBL theory, but highly relevant as an applied quality indicator for the integrity of physical qubit states—responds even more sensitively to parameter changes away from the deep localization limit. Expressed in the language of quantum information technology, it indicates strong ZZ coupling and the onset of ZZZ coupling already in regimes where the participation ratios are asymptomatic.

We note that the most recent experimental work has been strongly focused on the necessity to break the linkage between larger $T$ coupling and the appearance of ZZ coefficients. On the hardware side, ingenious new coupler schemes[21] show nearest-neighbor ZZ reduced to well below the danger level of 100 kHz of earlier work. It is further shown that control techniques, involving advanced refocusing strategies, also strongly diminish the effect of nearest-neighbor ZZ for a given $T$[22]. But our work provides a warning that these innovations may ultimately not be enough: all other couplings in the $\tau$-Hamiltonian hierarchy, including next neighbor ZZ and ZZZ, remain neither diagnosed nor ameliorated in the current experiments.

Third and finally, our study of precision-engineered qubit arrays has revealed a general structure which we believe is ubiquitous in weakly disordered transmon arrays: a restructuring of the "total Hilbert space" into subspaces which are weakly cross-correlated, but strongly ('chaotically') correlated within themselves. These hierarchies include spaces of fixed excitation number, or still further refined subspaces of these, distinguished by specific qubit permutation symmetries. Where this splintering occurs, a twofold task presents itself: first, identify the pattern of relevant spaces, and second apply the diagnostic tools discussed in earlier sections within these small spaces. Particular care must be exercised in cases where the relevant spaces overlap in energy. Absent considerable inter-space correlations one may then be tricked into the conclusion of Poissonian level statistics (localization!) where in fact wave functions are chaotically extended over the basis of a computationally relevant space. In

Supplementary Note 4, we detail all this intricate phenomenology, using the case study of the LASIQ engineered A-B-A-B pattern as an example illustrating these principles and the development of reliable predictions for the integrity of qubit wave functions.

What is the applied significance of these observations? One bottom line is that further reduction of the frequency variance may be dangerous. All transmon-based quantum technology operates in a tension field defined by the desire to optimally protect (detune) and efficiently operate (couple). There are different approaches to resolving this conflict of interests, scheme A "weak coupling/weak detuning" and scheme B "transient-incomplete coupling/strong detuning" defining two master strategies. Our study indicates that the A approach is more vulnerable to chaotic fluctuations. We go so far as to speculate that it may not sustain the generalization to larger and two-dimensionally interconnected array geometries required by more complex applications. However, regardless of what hardware is realized, the findings of this work indicate that the shadows of the chaotic phase are much longer than one might have hoped and that careful scrutiny of chaotic influences should be an integral part of future transmon device engineering.

## Methods

**Spectral statistics via Kullback–Leibler divergence.** To quantitatively analyze the spectral statistics, we look at the distribution of the ratios of adjacent level spacings $r_n = \Delta E_n / \Delta E_{n+1}$[23] (in order to avoid level unfolding) by comparing to what is expected for these ratios in Poisson or Wigner–Dyson statistics. This is often done via qualitative observations, such as focusing on the limit of $r \to 0$, where the distribution exhibits a maximum for Poisson statistics, but vanishes for Wigner–Dyson statistics (see, e.g., the insets of Fig. 3). However, a recent study[24] (of a Fock space localization transition) has shown that such inspections may trick one into false conclusions and that the Kullback–Leibler (KL) divergence[25] provides a far more reliable quantitative alternative. The KL divergence

$$D_{KL}(P \parallel Q) = \sum_k p_k \log\left(\frac{p_k}{q_k}\right), \tag{7}$$

defines an entropic measure quantifying the logarithmic difference between two distributions $P$ and $Q$. In our case, the $p_k$ are extracted from the numerical spectrum for a given set of parameters, while the $q_k$ follows one of the two principal spectral statistics considered here. Note that in Fig. 3 we plot $R_n = \min(r_n, 1/r_n)$ in order to restrict to the range [0, 1].

**Data collapse and phase transition.** A particularly consistent picture emerges if one performs a simple rescaling of the numerical data in both panels of Fig. 4. As shown in Fig. 8, the individual traces of both the KL divergence and the IPR for varying values of the Josephson energy $E_J$ (shown in the insets) all collapse onto one another when rescaling the coupling parameter $T \to T E_J^\mu$ with the exponent $\mu$ being the single free parameter. Such a data collapse is typically considered strong evidence for the existence of a *phase transition*, i.e., we can manifestly separate the MBL phase for small transmon couplings from a truly chaotic phase for sufficiently large couplings.

This also allows us to mark a Rubiconian line on our phase diagram (indicated by the green line in the top panel of Fig. 4) that should not be crossed in any quantum computation scheme, as all exquisitely prepared quantum information would be lost quickly upon entering the realm of quantum chaos lying beyond. The data collapse at a value $\mu \simeq 0.5$ follows from a simple argument: thinking of the wave functions as states living on a high dimensional lattice defined by the occupation number configurations $n = (n_1, n_2, \ldots, n_N)$ ($n_i = 0, 1$ for the computational subspace), individual occupation number states $n$ are connected to a large number of neighbors via the "hopping matrix elements", $t_{ij}$. Wave functions hybridize over a pair of configurations $n$, $m$, provided $|t_{ij}| \gtrsim |\Delta \epsilon_{nm}|$, where $\Delta \epsilon_{nm}$ is the energy difference between the two configurations in the limit $t \to 0$. Inspection of Eq. (2) shows that $\Delta \epsilon_{nm}$ depends on the $E_J$s as $E_{J_i}^{1/2} - E_{J_j}^{1/2} \sim E_J^{-1/2}(E_{J_i} - E_{J_j})$. In the analysis of Fig. 8, the random deviations are scaled such that $(E_{J_i} - E_{J_j}) \sim E_J^{1/2}$, such that $\Delta \epsilon_{nm}$ is effectively independent of $E_J$. However, $t_{ij} \sim T E_J^{1/2}$, indicating that $T E_J^{1/2}$ is the relevant scaling variable for the transition where the two parameters $T$ and $E_J$ are concerned.

**State identification for Walsh transform.** We find that there is a workable procedure for identifying computational states in the bosonic spectrum, which however starts to become problematic long before we reach the MBL-chaotic phase boundary. We adopt the following assignment procedure: at $T = 0$ all states have

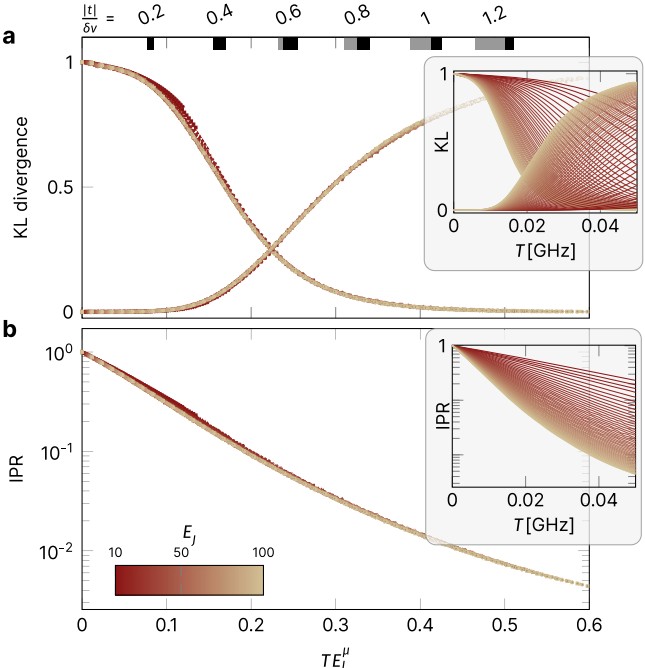

**Fig. 8 Data collapse.** The individual traces of the KL divergence and IPR, underlying the phase diagrams of Fig. 4 and shown in the two insets, are collapsed onto each other by rescaling the coupling parameter with respect to the Josephson energy as $T \to TE_J^\mu$ with an exponent $\mu \approx 0.54$. As $TE_J^\nu \sim |t|/\delta\nu \cdot E_J^{\mu-1/2}$, a $TE_J^\mu$ interval, whose lower (upper) bound is determined by the minimal (maximal) $E_J$ value, belongs to each $|t|/\delta\nu$ value. The areas shaded in black indicate the $TE_J^\mu$ range associated with the $|t|/\delta\nu$ label next to it. The upper boundary corresponds to $E_J = 100$ GHz. The lower boundary is obtained for the smallest $E_J$ for which data points at that particular $|t|/\delta\nu$ exist. The lower boundary of the gray-shaded interval corresponds to $E_J = 10$ GHz, the smallest Josephson energy considered in the data collapse.

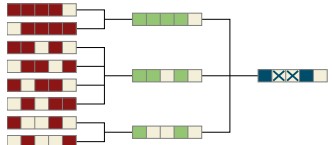

**Fig. 9 Color-coding of Walsh coefficients for a system of five-transmon qubits.** The first column shows the individual qubit assignments (light color = "0", red color = "1"), and the second and third columns indicate ways to average coefficients according to the enclosing brackets.

exact bosonic quantum numbers, so the $2^N$ eigenstates with bitstring label **b** (the Walsh transforms of the bitstrings in Eq. (5)) are immediately identified there. We increase $T$; as long as no near-crossings of energy levels occur, the labeling remains unchanged. We then find that the first near-crossings that occur have the character of isolated anticrossings with very small gaps. In this situation, we can confidently associate the label **b** with the diabatic state (i.e., the one that goes straight through the anticrossing). We show this by the coloring of Fig. 2c. Empirically, this identification procedure works through many anticrossings, up to about halfway to the phase transition; gradually, gaps become larger, and the identification of qubit states becomes more ambiguous (Fig. 2d). Naturally, in the chaotic phase, eigenstate thermalization says that there is no hope of consistently identifying any eigenstates as information-carrying multi-qubit states.

For clarity of visualization, we do not show all Walsh coefficients $c_\mathbf{b}$ in Fig. 5 and in Supplementary Fig. 2, but average over those with bitstring labels of an equal maximal distance of two 1's, as illustrated in Fig. 9 for a five-transmon chain with the coefficients of maximal distance four.

**Simulation parameters**. All simulation data shown in the main manuscript were calculated for $E_C = 0.25$ GHz. For scheme-A disorder, we use $\delta\nu \sim E_C/2$, resulting in

a Josephson energy spread $\delta E_J = \sqrt{E_C E_J/8}$. The Walsh-transform analysis was performed for $E_J = 12.5$ GHz.

**Classical chaos in transmon arrays**. We may incidentally remark that this work started from a project study[26] on *classical* chaos in the transmon system. Classically, the transmon Hamiltonian Eq. (1) describes a system of coupled mathematical pendula of mass $m = 1/8E_C$ and gravitational acceleration $g = 8E_C E_J$ (in units where $\hbar = 1$ and $\ell = 1$ (pendulum length)). Nonlinearly coupled pendula generally show a transition from integrable motion at low energies to hard chaos at high energies. (There are desktop gimmicks with just two coupled masses demonstrating the phenomenon.) The principal observation of the project was that already the classical two transmon Hamiltonian showed tendencies to chaos when excited to sufficiently high energies. The generalization to ten coupled oscillators made the situation worse, with Lyapunov exponents signaling uncontrollable dynamics for energies matching those of QC applications with 0 and 1 qubit states, and at time scales way below typical coherence times.

## Data availability
The data that support the findings of this study are available from the corresponding author upon reasonable request.

## Code availability
The code used to generate the data used in this study is available from the corresponding author upon reasonable request.

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

## Acknowledgements
We thank S. Börner for collaboration on an initial project[26] studying incipient chaos in classical transmon systems. We acknowledge partial support from the Deutsche Forschungsgemeinschaft (DFG) under Germany's Excellence Strategy Cluster of Excellence Matter and Light for Quantum Computing (ML4Q) EXC 2004/1 390534769 (E.V., S.T., A.A., and D.P.D.) and within the CRC network TR 183 (project grant 277101999) as part of projects A04 and C05 (C.B., S.T., and A.A.). The numerical simulations were performed on the CHEOPS cluster at RRZK Cologne and the JUWELS cluster at the Forschungszentrum Jülich.

## Author contributions
S.T., A.A., and D.P.D. conceived the project and coordinated the research. C.B. and E.V. performed all the simulations. All authors discussed and analyzed the results and contributed to writing the manuscript.

## Funding

## Competing interests
The authors declare no competing interests.
