## [Peer Review File · Nature Communications]

REVIEWER COMMENTS

Reviewer #1 (Remarks to the Author):

The manuscript “Transmon platform for quantum computing challenged by chaotic fluctuations” merges two intuitively distinct fields: many-body localization and quantum processor engineering for quantum computation. The main focus is on the question what is the right amount of disorder in transmon devices compared to their coupling rate in a geometry and parameter regime of typical superconducting quantum processors.

Many-body localization has taught that whenever disorder is large enough with respect to hopping rate it is sensible to consider the system made of quasi-local subsystems despite particle-particle interactions (many-body interaction). Then information exchange and build-up of entanglement between other quasi-local subsystems is logarithmically slow. In this regime, it is sensible to attempt quantum gates with local controls and exercise quantum computing in terms of the quasi-local subsystems. When disorder is not large enough with respect to hopping rates, the protection of quasi-local subsystems by localization crumbles. Information and entanglement spreads fast and chaotically throughout the physical volume of the system (as well as the whole Hilbert space). In this regime, the concept of quantum computing with locally controlled quantum gates is not sensible.

The strongest merit of the manuscript is that it really brings the concepts, methods, and results of many-body localization in interplay with state-of-the-art superconducting quantum processor designs and devices, and attempts to resolve how close the processors are from the transition to the chaotic fluctuations with actual physical parameters. The manuscript is quite a technical one but it is evident that authors have tried to tailor language and notations to satisfy specialists from both communities and those in between, which is always respectable. Please see below some suggestions for improving the readability.

Quantum computing and chaotic fluctuations are both dynamical phenomena. A gate-based quantum algorithm is a temporal series of quantum gates and measurements, where it is essential to know how does the quantum state evolves in time. Likewise, chaotic fluctuations, Anderson localization and many-body localization are most accurately distinguished in dynamical quantities (temporal evolution of entanglement entropy, correlations, number fluctuations). The methods of the manuscript analyze only energy levels and wavefunctions. My main concern is related to the question whether or not the methods and drawn conclusions are indeed reflected into dynamics relevant for quantum computations. No dynamical quantity is demonstrated in the manuscript. Please see below more detailed questions.

Without any hesitation it is easy to say that the manuscript is well-written and is of high scientific value. It reports a very interesting theoretical and computational results of a demanding scientific question. However, I feel sorry that my main concern on implications to actual quantum computation dynamics precludes me immediately recommending publication in Nature Communications.

Major points:

Question: Are spectral and wave function statistics and Walsh transformation results computed only for “the computational states” , i.e. the states that are highlighted with red in Fig 1(c)-(d)? If yes, then the comment/questions (1) & (2) are relevant and need answering.

(1) I understand that it is possible to consider energy level statistics just for a subset of states [e.g. just for states that can be identified as computational states within an energy window]. However, my question is that how is the statistics of subset energy levels and wavefunctions related to the actual dynamics in the system? Isn't it important to include also all the other states where local excitation number can be higher than 1? For example, does the analysis include processes that cause leaking out from the computation subspace, for example, $|0101010101\rangle \rightarrow |0200010101\rangle$?

(2). What is the logic in choosing the energy window in the middle of the total energy spectrum of the Hamiltonian (1)? It is quite an easy task to imagine a situation where the middle energy lands in the gap of no energy levels at all in Fig 1(a) [i.e. in the gap between 4 and 5 total excitations in the system].

(3). Related to the question (2): Based on the Fig 1(a), one can argue that the total excitation number is a well conserved quantity [the gaps between the different excitation manifolds are well resolvable]. Then, with respect to many-body localization literature, it would be more natural to consider just a manifold of a fixed total excitation number and take energy levels in mid-spectrum of that manifold.

(4). What is the logic of choosing to study the states with exactly $N/2$ bit flips? I know that the numerics becomes the harder the more excitations are considered. In quantum computation, one is not limited to states at half-filling and all multi-qubit states are equally important. Both $|0101010101\rangle$ and $|1111111111\rangle$ are equally good quantum computation states. I would guess that at high filling, the system would become more prone for delocalization and chaotic fluctuations?

Minor points:

1. What is the width of the energy level window where the energy levels are picked?

2. Figures 4 and Figures 5 bc are hard to understand. Why do you need to have separate labeling (top and bottom) for $T=50$ MHz and $T=2$ MHz in Fig 4(a) [and likewise in Fig 5b]? I guess the reason is that the values $|c_b|$ are ordered by their magnitude in both cases, couldn't it be done just with a set of labels? Furthermore, it would be nice if you could explain the notation of the labeling with numeric bit strings. Actually, why not use just numeric bit strings?
3. It would be nice if you could give the value or the range of disorder strength used in each figure in units with respect to T , I mean the value or range of $\Delta E_J/T$ and $\Delta \nu / |t|$. The raw values are given in Methods section but the scaled values are important quantities to know when comparing the current results with the existing many-body localization literature results.
4. Define what $\langle i,j \rangle$ means in Eqs. (1) and (2).
5. In sentence "This scale is much larger than the particle hopping strength, which for the same parameter set is about $J \approx 6$ MHz" on page 2, you use J as a hopping strength, but in the Hamiltonian (2) it is t_{ij} . Or does it refer to T in Eq. (1)?
6. What is the motivation to use the full Hamiltonian (1) instead of the Bose-Hubbard (2)? Maybe could be explained in one sentence.
7. What is the basis in which the inverse participation ratio is computed?

Reviewer #2 (Remarks to the Author):

In the manuscript "Transmon platform for quantum computing challenged by chaotic fluctuations" the authors investigate a many-body model described by an effective Hamiltonian, which incorporates second-order one-site nonlinearity. The model also includes disorder which enters it through variations of the amplitudes of two terms, linear on-site energy and hopping. The model Hamiltonian approximates the Hamiltonian describing an array of interacting transmon qubits, where the source of the disorder is the variations of the Josephson-junction charging energy. The key object of the investigation is the transition between many-body ergodic phase ('chaos') and many-body localization (MBL).

The authors use a spectrum of methods to quantify this transition and resolve the parameter dependencies 'chaos-MBL'. The strengths of variations of charging energy and of transmon coupling are two key parameters used to control the transition between chaotic and MBL phases. By using different values of the disorder strength (quantified with the standard deviation) of the Josephson energy, typical to the IBM and Delft/Google transmon platforms, the authors demonstrate that in the first case the model exhibits regime close to the chaos-MBL transition (from the MBL side), while in the second case it appears to be deeply in the MBL regime.

The ms presents new interesting results, it is very well written, and the presentation is appropriately detailed.

I also consider the ms to be 'interdisciplinary' since it addresses a topic located on the interface of two fields, quantum computing technology and MBL. As such, it bridges these two fields in a nice way and invites other researchers – from these two and other fields – to investigate the topic further. The results presented in the ms are of interest to the broad readership of Nature Com, especially in the context of current trends such as the role of disorder in the performance of the current NISQ computing platforms, MBL regimes in JJ-related models (more complex than the standard spin models), and quantum control.

.

I am happy to recommend publication in Nature Com.

Reviewer #3 (Remarks to the Author):

While superconducting qubit arrays have previously been considered as a platform for simulation of many localization (MBL), Berke and authors turn this around and discuss the implications of MBL physics for building transmon based quantum processors. Applying tools from MBL theory to the standard coupled transmon Hamiltonian – the authors study parameter regimes where the system breaks into a chaotic phase, where the computational levels can no longer be identified unambiguously. These authors analyze various cases: operating regimes of prevalent superconducting quantum processors from IBM/Google/Delft and varying qubit lattices and frequency arrangements. This is certainly a very novel piece of work, and a unique analysis of superconducting processor operating regimes.

In my opinion, a major drawback of this manuscript lies in the presentation, which I understand is a challenge given the scope of the work. The paper combines concepts of superconducting qubit physics with many body localization, which are both areas of rich study that have previously not shared a natural overlap in their respective communities. In this context, I would strongly encourage the authors to be more pedantic in their introductory discussions and introduce terminology through the manuscript in a manner that offers some background to both communities. For instance, a lot of literature that is likely unfamiliar to superconducting qubit physicists is introduced

at the start of section on Diagnostics. Although it gets detailed later in the Methods, the unfamiliar reader is a little clueless without relevant references at that point. Having that text integrated with the main text should improve readability. Along these lines, in Figure 2, there are many labels that have not been previously introduced or defined. (D_{KL}, r_n, R_n, p). Also, all figures should have a more detailed description of the Hamiltonian parameters employed (or a reference to the Methods section) – for instance, “small site-to-site variations characteristic of the IBM design” on Figure 2 is not ideal for readers looking to reproduce these simulations.

Towards enabling easier readership by a broader community, instead of referring to the regimes of operation as IBM/Delft/Google, I think that it will help to summarize upfront the considered regimes of operation for what they physically are. Example:

Type A: fixed T couplings with small frequency spread

Type B: fixed T couplings with larger, tunable frequency spread

Type C: tunable T couplings with smaller, tunable frequency spread

There are several groups that employ operating regimes similar to A/B/C, but this also helps keep track for readers not intimately familiar with the different variants of superconducting quantum processors. Also it might also help to summarize the δ_v and T associated with these cases. The discussions section does some version of this, but it would be helpful to define these regimes at the start.

As pointed by the authors, the trade-off between increasing the qubit-qubit coupling parameter T and errors from local two-body terms (ZZ) has been long studied in superconducting qubit literature as a challenge in fixed-coupling architectures. However, this discussion needs to be much more nuanced in light of recent advances from the IBM group addressing this tradeoff (arXiv:2106.00675, arXiv:2011.07050), and its implication on the limits to increasing T.

In the discussion of Fig 5: “Although definitely at a lower level, the higher-order terms of the τ - Hamiltonian are still present, with the ZZ danger threshold (dashed line) still reached. This occurs only at around $T = 9$ MHz, about three times larger than for the natural-disorder parameters. But dangerous MBL phenomenology is still lurking in the wings, even in this much more favorable setting. “

Why is this ZZ danger threshold (somewhat arbitrarily defined here at 100kHz) relevant for the case of tunable couplers?

As the authors also point out, the A-B-A frequency arrangement is an oversimplification but serves to highlight some interesting features in their analysis in the limit of very small disorder. However, I

don't follow this conclusion by the authors: "What is the applied significance of these observations? One is that one should not aim to reduce the frequency variance much farther. "

Why is the reduced variance a problem if you want to use a more precise but more complex frequency arrangement of qubits?

The authors consider 3 different indicators of chaotic fluctuations in MBL theory, but discuss that the onset of chaotic behavior is seen differently in their parameter sweeps. Can the authors comment on whether there is any understanding of why these differences arise or why certain diagnostics are not as sensitive? In the future, which diagnostic would be the go-to for such an analysis?

The analysis presented in this work is undeniably unique and an interesting take on the problem, but I am struggling to find a novel insight in superconducting quantum processor design that has not been arrived at already in the traditional approach, and find some of the conclusions questionable. For instance, in the language of the superconducting quantum computing community, increasing coupling T typically leads to large static ZZ errors, hence, one should either operate at weak T fixed couplers or make tunable T couplers. Along these lines, how is it meaningful to make a comparison of the different regimes across the same extent of T , and then conclude that the IBM strategy is more sensitive to these chaotic fluctuations? Furthermore, many of the design considerations in practice are far more nuanced compared to the discussions in this work -- frequency arrangement, qubit lattice and coupling vs coherence. While some of these are acknowledged by the authors in the main body, they seem to be ignored in the rather strong concluding remarks.

Minor comments:

- Figure 5b, label for contour missing (I guess it is $\frac{1}{2}$?)
- Discussion on frequency spreads switches between ΔE_J and Δv – I would suggest sticking to one, for consistency when comparing regimes.

Response to the referee reports

We thank the three referees of our manuscript for their detailed and informed feedback. We appreciate the substantial amount of effort required to work through the paper and provide detailed reports. While these naturally differ in some aspects, we see major recurrent patterns, which played a crucial role in preparing a substantially changed and restructured manuscript:

- All three reports seem to appreciate the timeliness of an interdisciplinary effort in applying concepts of localization theory to the engineering effort of building better quantum devices. The principal paradigm underlying our work — the Janus-faced nature of disorder as both a foe and a critically required resource — remains undisputed. Encouraging statements include “The strongest merit of the manuscript is that it really brings the concepts, methods, and results of many-body localization in interplay with state-of-the-art superconducting quantum processor designs and devices” (referee #1), “... it bridges these two fields in a nice way and invites other researchers — from these two and other fields — to investigate the topic further.” (referee #2), or “while superconducting qubit arrays have previously been considered as a platform for simulation of many localization (MBL), Berke and authors turn this around ... this is certainly a very novel piece of work, and a unique analysis.” (referee #3).
- All referees point out the “interdisciplinary” nature of our work. However, we receive mixed signals concerning the presentation. While all referees appear to acknowledge the challenges of talking to different communities, referee #1 considers the manuscript as “quite a technical one” while #3 sees deficits in the presentation and encourages us to be “a bit more pedantic” (which we translate as a request for more detailed discussion). In fact, the comments of the referees reflect the presence of a thin line we felt we were walking since the beginnings of the project: models considered in most MBL studies — spin chains, mostly — are *way* simpler than even basic transmon arrays. Any study aiming to be of applied relevance must take the fine print of transmon qubit design into account. The necessity to be specific is illustrated by our results for the LASIQ engineered array in the paper, and even more so by the answer to one of the specific technical questions of referee #3. In writing the manuscript, we had to strike a balance between being too vague and getting bogged down in details. However, prompted by the input of the referees we have significantly restructured the text (see list of changes below) and hope that it is now both more pedagogical and informative.
- Finally, there remains the question of the concrete relevancy of our findings. The most serious objection is raised by referee #1 “my main concern on implications to actual quantum computation dynamics precludes me immediately recommending publication in Nature Communications”, and #3 “but I am struggling to find a novel insight in superconducting quantum processor design that has not been arrived at already in the traditional approach”. We do not have a one–does–it–all argument that could dispel these concerns. Instead, it may be best to identify points on which we perhaps can all agree: We translate the “traditional approach” of referee #3 as one where the focus is on blocking qubit neighbor resonances (ZZ-couplings). These are of course the most dangerous ones, they must and can be avoided. On the other hand, the theory of MBL, since its beginning, has been about the influence of more remote correlations in Fock spaces, and their influence on the structure of many body states. These are (exponentially in distance) weaker than ZZ; however, they are also (exponentially) more numerous. MBL starts with the observation that these two principles largely balance each other, making non local correlations crucially important. The question is, how relevant this principle is for concrete processor design.

One indication is provided by the trivial observation that the limit of a clean array patterned into precision engineered “unit cells” must be avoided. Its many body eigenstates would be extended, due to principles outside the ZZ-paradigm. Our analysis of the relatively simple ABAB pattern shows, we believe, that this threat is not an academic one. We have found that for too low disorder, the (computationally relevant) Hilbert space fragments into subspaces of states extended over multiple AB cells with intra-subspace chaotic correlations. These state structures

are treacherous in that standard indicators of localization (spectral statistics) may trick one into the conclusion that the situation is safe, while more refined ones (Walsh transform beyond ZZ-level) signal a danger. Naturally the situation gets more involved when network geometries beyond strictly one dimensional, or more involved patterns of qubit engineering are considered. However all experience of many body localization theory points at enhanced tendencies to delocalized regimes in higher connected geometries. On this background, yes, we believe that the physics discussed in the paper is relevant, including from an engineering perspective.

At this point “all” we can offer to corroborate this belief are quantitative predictions for non-local correlations of qubit states in what we hope are reasonably realistic model proxies for existing devices. Probably only time (translate, experimental studies of concrete devices) will decide if we got this right. At this point, we are asking to give this MBL-biased perspective a chance to reach a broad and interdisciplinary readership. (In response to referee #3’s concrete concerns, we have run some specific diagnostics and included its results, later in this reply letter.)

The comments of the referees have motivated us to substantially revise the paper. Specifically,

- We now open the paper with a discussion of the structure, and the disorder distribution of current generations of IBM processors (for which this data is publicly available on the web). We hope that this early specification will give readers a more vivid impression of the systems we are modelling.
- Following a suggestion of referee #3 we distinguish between different types of disorder distributions, calling them scheme A and scheme B, respectively (rather than referring to IBM- and Google-disorder as in the previous submission).
- Of our three diagnostics, the Walsh transform is the most sophisticated, but also the most challenging to conceptualize. We have now added a more pedagogical discussion of the interpretation of Walsh transform data.
- We have realized that of the latest generations of IBM chips, only few use LASIQ engineered alternate frequency patterns. We have therefore moved the detailed and comparatively involved analysis of the weakly disordered ABAB patterned transmon array to the supplemental material, retaining only a summary discussion in the main text.

We feel that with this major revision of our manuscript which reflects the informed input from all three reviewers, our work should now be cleared for publication in Nature Communications.

Yours sincerely,

Christoph Berke (for all authors)

Reviewer #1 (Remarks to the Author):

The manuscript “Transmon platform for quantum computing challenged by chaotic fluctuations” merges two intuitively distinct fields: many-body localization and quantum processor engineering for quantum computation. The main focus is on the question what is the right amount of disorder in transmon devices compared to their coupling rate in a geometry and parameter regime of typical superconducting quantum processors.

Many-body localization has taught that whenever disorder is large enough with respect to hopping rate it is sensible to consider the system made of quasi-local subsystems despite particle-particle interactions (many-body interaction). Then information exchange and build-up of entanglement between other quasi-local subsystems is logarithmically slow. In this regime, it is sensible to attempt quantum gates with local controls and exercise quantum computing in terms of the quasi-local subsystems. When disorder is not large enough with respect to hopping rates, the protection of quasi-local subsystems by localization crumbles. Information and entanglement spreads fast and chaotically throughout the physical volume of the system (as well as the whole Hilbert space). In this regime, the concept of quantum computing with locally controlled quantum gates is not sensible.

The strongest merit of the manuscript is that it really brings the concepts, methods, and results of many-body localization in interplay with state-of-the-art superconducting quantum processor designs and devices, and attempts to resolve how close the processors are from the transition to the chaotic fluctuations with actual physical parameters.

The manuscript is quite a technical one but it is evident that authors have tried to tailor language and notations to satisfy specialists from both communities and those in between, which is always respectable.

We thank the referee for the positive remarks!

Please see below some suggestions for improving the readability.

Quantum computing and chaotic fluctuations are both dynamical phenomena. A gate-based quantum algorithm is a temporal series of quantum gates and measurements, where it is essential to know how does the quantum state evolves in time. Likewise, chaotic fluctuations, Anderson localization and many-body localization are most accurately distinguished in dynamical quantities (temporal evolution of entanglement entropy, correlations, number fluctuations). The methods of the manuscript analyze only energy levels and wavefunctions. My main concern is related to the question whether or not the methods and drawn conclusions are indeed reflected into dynamics relevant for quantum computations. No dynamical quantity is demonstrated in the manuscript. Please see below more detailed questions.

We share the referee’s point of view that the physics of MBL finds its most differentiated expression in dynamical phenomena. For example, logarithmic entanglement growth in time is a unique signature of MBL not observed in single particle Anderson localization. And of course, a quantum processor is dynamically operated, too. At the same time, the analysis of dynamics even for very simple MBL systems is more demanding than that of wave functions and spectra. In fact, there is a complexity hierarchy. For example, logarithmic entanglement growth is the dynamical manifestation of the non-local wave function correlations encoded in the Walsh transform. We therefore believe that the study of dynamics defines a natural next step, building on the solid understanding of wave functions and spectra; we are currently taking first steps in this direction. However, we also believe that the inclusion of dynamics would have blown the limits of an already complex paper.

Without any hesitation it is easy to say that the manuscript is well-written and is of high scientific value. It reports a very interesting theoretical and computational results of a demanding scientific question. However, I feel sorry that my main concern on implications to actual quantum computation dynamics precludes me immediately recommending publication in Nature Communications.

We have tried to express our view concerning the referee’s reservation in the general preamble to this reply letter.

Major points:

Question: Are spectral and wave function statistics and Walsh transformation results computed only for “the computational states” , i.e. the states that are highlighted with red in Fig 1(c)-(d)?

The answer is that, except for the Walsh-transform analysis, we did in fact take all states in the relevant window, not just the computational ones.

If yes, then the comment/questions (1) & (2) are relevant and need answering.

(1) I understand that it is possible to consider energy level statistics just for a subset of states [e.g. just for states that can be identified as computational states within an energy window].

However, my question is that how is the statistics of subset energy levels and wavefunctions related to the actual dynamics in the system? Isn't it important to include also all the other states where local excitation number can be higher than 1? For example, does the analysis include processes that cause leaking out from the computation subspace, for example, $|01010101\rangle \rightarrow |0200010101\rangle$?

We agree with the referee that the full Hilbert space is essential to the understanding of the quantum statistical mechanics of our system. Accordingly, our analysis of eigenenergy statistics and inverse participation ratios takes all, computational and non-computational states, into account. Only in the more refined Walsh transform, we have restricted ourselves to computational subspaces. The rationale here is that Walsh diagnostics makes sense only in cases where computational states can be isolated from others via effective selection rules. (If this were not an option, the whole program of protected quantum computation would be compromised.)

(2) What is the logic in choosing the energy window in the middle of the total energy spectrum of the Hamiltonian (1)? It is quite an easy task to imagine a situation where the middle energy lands in the gap of no energy levels at all in Fig 1(a) [i.e. in the gap between 4 and 5 total excitations in the system].

(3) Related to the question (2): Based on the Fig 1(a), one can argue that the total excitation number is a well conserved quantity [the gaps between the different excitation manifolds are well resolvable]. Then, with respect to many-body localization literature, it would be more natural to consider just a manifold of a fixed total excitation number and take energy levels in mid-spectrum of that manifold.

These are excellent points which warrant a detailed answer: The total excitation number N_{ex} is indeed a good quantum number that can be assigned to the lower bundles for ‘scheme A’ disorder: manifolds of fixed N_{ex} are energetically separated, and in addition, the interaction does not couple them to adjacent bundles $N_{\text{ex}} \pm 1$ (or more generally $N_{\text{ex}} \pm k$, where k is odd).

For the results presented in the manuscript, we do not fix an absolute energy window, but we average over all states / levels taken from a fixed N_{ex} manifold for each disorder realization. For the results shown in Fig. 3,4,6,8 of the manuscript, $N_{\text{ex}} = 5$, such that for each disorder realization, 2002 (10 transmons), 1287 (9), and 462 (7) states are taken into account. As shown in Fig. R1 (see below), the KL divergence is independent of whether the target energies are obtained from a Hamiltonian restricted to the desired sector of N_{ex} , or whether they are a part of the spectrum of a much larger Hamiltonian, whose basis also include states of different N_{ex} .

For ‘scheme B’ disorder, there are no clearly resolved manifolds of fixed N_{ex} . We conducted our analysis repeatedly and selected states using one of the following recipes (described here for a 10 transmon chain): (i) Choose the states / energies $\epsilon_i, |\psi_i\rangle$, $i = 1001, \dots, 3003$ (these states form the $N_{\text{ex}} = 5$ manifold for ‘scheme A’ disorder). (ii) Choose ~ 2000 states centered around the mean energy of the $N_{\text{ex}} = 5$ basis states at $T = 0$. In both cases, the selected states dominantly belong to $N_{\text{ex}} = 5$,

Figure R1 – KL divergence for the $N_{\text{ex}} = 5$ manifold of a 10 transmon chain. The Hamiltonian (a) includes basis states of $N_{\text{ex}} = 0, \dots, 7$ ($\dim \mathcal{H} \approx 20000$) and (b) is restricted to the $N_{\text{ex}} = 5$ manifold ($\dim \mathcal{H} = 2002$). Except for the coarser T - E_J grid in (a), the results are essentially identical.

but states of higher and (fewer) states of lower N_{ex} are also included. (iii) Restrict the Hamiltonian to the $N_{\text{ex}} = 5$ subspace, even if this neglects states which are fully intermingled with $N_{\text{ex}} = 5$ states. In the T - E_J parameter regions under consideration, all approaches give nearly identical results. In particular, Fig. 2(a) of the Supplementary Information – calculated using the approach (iii) – does not change for (i) and (ii), and in Fig. 2(b), the line where $\text{IPR}=1/2$ moves towards the lower left corner by a barely discernible amount.

Given that for ‘scheme A’ disorder, N_{ex} is conserved, we agree with the referees reasoning that an alternative approach often considered in MBL literature is to pick energy levels from the band center of a fixed N_{ex} manifold, e.g., levels near a fixed normalized energy $\epsilon = (E - E_{\text{min}}) / (E_{\text{max}} - E_{\text{min}})$, where E_{max} (E_{min}) is the largest (smallest) energy in the desired sector. Fig. R2(b) below shows the ϵ -resolved KL divergence. In (a), the total number of states (blue) is shown as a function of ϵ (for $T = 0$). The location of the transition between chaos and MBL depends on ϵ (‘many-body mobility edge’). If comparing this to (a), one sees that the higher the DOS, the more prone the system is to chaos. Note that the computational states, whose number is shown in green in (a), sit primarily at the upper end of the spectrum, in the ϵ range that is particularly chaotic. In (c) (bottom panel), we reevaluated the data shown in Fig. 4(a) of the manuscript, but levels are now taken from an interval of length $\Delta\epsilon = 0.05$ around the maximal DOS. For comparison, the top panel shows the KL divergence obtained for the whole N_{ex} bundle. The solid (dashed) line indicates where the KL divergence calculated for the small interval (the full bundle) crosses $1/2$. When rescaling the axis $T \rightarrow TE_J^\mu$ – as shown in (d) – the individual traces of the KL divergence collapse for $\mu \approx 0.54$, the same exponent used in Fig. 8 of the manuscript. For comparison, the blue line shows the collapsed data for the full bundle.

Note that, even for ‘scheme A’ disorder, bundles start to overlap above a few excitations and that for larger transmon arrays, unbound states might also intermingle already at low filling fractions. The bottom line is that the choices that we have made in the manuscript give a robust picture of the situation with energy level statistics.

(4) What is the logic of choosing to study the states with exactly $N/2$ bit flips? I know that the numerics becomes the harder the more excitations are considered. In quantum computation, one is not limited to states at half-filling and all multi-qubit states are equally important. Both $|01010101\rangle$ and $|11111111\rangle$ are equally good quantum computation states. I would guess that at high filling, the system would become more prone for delocalization and chaotic fluctuations?

The states with $N/2$ bit flips are obviously the most numerous among the computational states, and thus can be viewed as most representative. We agree that all states, including $|11111111\rangle$, are important for quantum computation. This is an important function of the Walsh-transform metric, as its parameter extraction is sensitive to all excitation bands.

In Fig. R3(a), we show results for the KL divergence for $N = 10$ and various N_{ex} ranging from 2 to 7. For each disorder realization, all levels of a respective bundle are used for the R_n distribution.

Figure R2 – (a) Number of states (blue) and number of computational states (green) as a function of the normalized energy ϵ for $N_{\text{ex}} = 6$ in a 12 transmons chain (disorder average at $T = 0$). (b) ϵ -resolved KL divergence ($E_J = 44$ GHz, $E_C = 250$ MHz) for ‘scheme A’ disorder, showing that the transition from MBL to chaos depends on ϵ . (c) KL divergence for $N_{\text{ex}} = 5$ and a 10 transmons chain, ‘scheme A’ disorder. The upper panel is identical to Fig. 3(a) in the paper. For the lower panel, only states from an ϵ interval of width $\Delta\epsilon = 0.05$ around the maximal DOS are considered. The solid (dashed) line indicate where the KL divergence crosses 1/2 in the lower (upper) panel. (d) Data collapse for the KL divergence based on states near the maximal DOS. For comparison, the blue line shows the collapsed data for the full bundle

As anticipated by the referee, the system becomes more prone to chaotic fluctuations as the filling is increased. This effect can be seen more clearly in (b), where only states near the maximal DOS are considered. Focusing on generic states, our stability analysis, thus, is a conservative one. Had we used the integrity of all states as a criterion, the conclusions on device stability would have been less optimistic.

Figure R3 – KL divergence for a 10 transmon chain and various total excitation numbers $N_{\text{ex}} = 2, \dots, 7$ for $E_J = 44$ GHz, $E_C = 250$ MHz. In (a), the KL divergence is calculated by averaging over all levels of the respective bundle (e.g. 55 levels for $N_{\text{ex}} = 2$ and 11440 levels for $N_{\text{ex}} = 7$). In (b), only the levels in an interval of length $\Delta\epsilon = 0.05$ ($\Delta\epsilon = 0.1$ for $N_{\text{ex}} = 3, 4$) around the maximal DOS are considered. Both calculations show the same tendency of a stronger increasing KL divergence for higher total excitation numbers.

Minor points:

1. *What is the width of the energy level window where the energy levels are picked?*

See the detailed answer to major point (3) above.

2. *Figures 4 and Figures 5 bc are hard to understand. Why do you need to have separate labeling (top and bottom) for $T=50$ MHz and $T=2$ MHz in Fig 4(a) [and likewise in Fig 5b]? I guess the reason is that the values $|c_b|$ are ordered by their magnitude in both cases, couldn't it be done just with a set of labels? Furthermore, it would be nice if you could explain the notation of the labeling with numeric bit strings. Actually, why not use just numeric bit strings?*

We have revised and simplified these figures. Excessive contents has been removed and an enhanced caption now explains in hopefully pedagogical terms the bit-string graphic.

3. *It would be nice if you could give the value or the range of disorder strength used in each figure in units with respect to T , I mean the value or range of $\delta E_J/T$ and $\delta\nu/|t|$. The raw values are given in Methods section but the scaled values are important quantities to know when comparing the current results with the existing many-body localization literature results.*

We have added information on the value / range of δE_J to all relevant figure captions. However, the dimensionless quantities $|t|/\delta\nu$ and $T/\delta E_J$ change in a non-trivial way in the T - E_J diagrams: $|t|/\delta\nu$ and $T/\delta E_J$ change linearly along the T axis, and $\sim \sqrt{E_J}$, respectively $\sim 1/\sqrt{E_J}$ along the E_J axis. Lines of constant $|t|/\delta\nu$ ($T/\delta E_J$) have the form $E_J \sim 1/T^2$ ($E_J \sim T^2$). The highest values are $\sqrt{2}$ in the upper right corner and $\approx 1/8$ in the lower right corner (for scheme-A parameters). In our opinion, adding this information does not bring additional clarity. We feel that the data collapse section is the best place to specify absolute values of the relevant scaling variable $|t|/\delta\nu$. We therefore updated Fig. 8 that now includes an additional $|t|/\delta\nu$ axis.

4. *Define what means in Eqs. (1) and (2).*

Unfortunately, we could not understand the request here; We hope that all relevant definitions are given after these equations.

5. *In sentence “This scale is much larger than the particle hopping strength, which for the same parameter set is about $J \approx 6$ MHz” on page 2, you use J as a hopping strength, but in the Hamiltonian (2) it is t_{ij} . Or does it refer to T in Eq. (1)?*

This is indeed a typo — thanks for spotting it —, and we have corrected it to $|t_{ij}|$.

6. *What is the motivation to use the full Hamiltonian (1) instead of the Bose-Hubbard (2)? Maybe could be explained in one sentence.*

Numerically, (1) and (2) are equally easy to simulate. However, (2) is a low energy approximation, and we wanted to be sure that, particularly for the highest energy bands and largest T , the Bose-Hubbard approximation does not generate errors. Eq. (2) is introduced to define various auxiliary parameters, relate to previous work, and to the intuition of people working in MBL. However, later we state that the bulk of the paper works with the full model (1).

7. *What is the basis in which the inverse participation ratio is computed?*

We do this in the Fock basis, as is now remarked.

Reviewer #2 (Remarks to the Author)

In the manuscript “Transmon platform for quantum computing challenged by chaotic fluctuations” the authors investigate a many-body model described by an effective Hamiltonian, which incorporates second-order one-site nonlinearity. The model also includes disorder which enters it through variations of the amplitudes of two terms, linear on-site energy and hopping. The model Hamiltonian approximates the Hamiltonian describing an array of interacting transmon qubits, where the source of the disorder is the variations of the Josephson-junction charging energy. The key object of the investigation is the transition between many-body ergodic phase (‘chaos’) and many-body localization (MBL).

The authors use a spectrum of methods to quantify this transition and resolve the parameter dependencies ‘chaos-MBL’. The strengths of variations of charging energy and of transmon coupling are two key parameters used to control the transition between chaotic and MBL phases. By using different values of the disorder strength (quantified with the standard deviation) of the Josephson energy, typical to the IBM and Delft/Google transform platforms, the authors demonstrate that in the first case the model exhibits regime close to the chaos-MBL transition (from the MBL side), while in the second case it appears to be deeply in the MBL regime.

The ms presents new interesting results, it is very well written, and the presentation is appropriately detailed. I also consider the ms to be ‘interdisciplinary’ since it addresses a topic located on the interface of two fields, quantum computing technology and MBL. As such, it bridges these two fields in a nice way and invites other researchers – from these two and other fields – to investigate the topic further. The results presented in the ms are of interest to the broad readership of Nature Com, especially in the context of current trends such as the role of disorder in the performance of the current NISQ computing platforms, MBL regimes in JJ-related models (more complex than the standard spin models), and quantum control.

I am happy to recommend publication in Nature Com.

We thank the referee for their positive evaluation and recommendation for publication!

Reviewer #3 (Remarks to the Author)

While superconducting qubit arrays have previously been considered previously as a platform for simulation of many localization (MBL), Berke and authors turn this around and discuss the implications of MBL physics for building transmon based quantum processors. Applying tools from MBL theory to the standard coupled transmon Hamiltonian – the authors study parameter regimes where the system breaks into a chaotic phase, where the computational levels can no longer be identified unambiguously. These authors analyze various cases: operating regimes of prevalent superconducting quantum processors from IBM/Google/Delft and varying qubit lattices and frequency arrangements. This is certainly a very novel piece of work, and a unique analysis of superconducting processor operating regimes.

In my opinion, a major drawback of this manuscript lies in the presentation, which I understand is a challenge given the scope of the work. The paper combines concepts of superconducting qubit physics with many body localization, which are both areas of rich study that have previously not shared a natural overlap in their respective communities. In this context, I would strongly encourage the authors to be a more pedantic in their introductory discussions and introduce terminology through the manuscript in a manner that offers some background to both communities. For instance, a lot of literature that is likely unfamiliar to superconducting qubit physicists is introduced at the start of section on Diagnostics. Although it gets detailed later in the Methods,

The Diagnostics and the Methods sections are now more strongly interlinked. Many references have been moved up to the Diagnostics section in the process.

The unfamiliar reader is a little clueless without relevant references at that point. Having that text integrated with the main text should improve readability. Along these lines, in Figure 2, there are many labels that have not been previously introduced or defined. (D_{KL}, r_n, R_n, p).

We thank the reviewer for this comment. We have adopted the figure (replacing p by the word “density” and its caption to include a pointer to the definition of the Kullback-Leibler divergence D_{KL}). The discussion of the ratios r_n and R_n is a rather technical one and has been pushed to the Methods section.

Also, all figures should have a more detailed description of the Hamiltonian parameters employed (or a reference to the Methods section) – for instance, “small site-to-site variations characteristic of the IBM design” on Figure 2 is not ideal for readers looking to reproduce these simulations.

We hope that the newly added Fig. 1 deals with all these inadequacies related to experimental parameters and the motivation for scheme A. Figures now generally have much more extensive captions.

Towards enabling easier readership by a broader community, instead of referring to the regimes of operation as IBM/Delft/Google, I think that it will help to summarize upfront the considered regimes of operation for what they physically are. Example: Type A: fixed T couplings with small frequency spread Type B: fixed T couplings with larger, tunable frequency spread Type C: tunable T couplings with smaller, tunable frequency spread

This is a very good suggestion which we have gladly taken up. However, we have realized that a simpler categorization with only two distinctions, ‘scheme A’ and ‘scheme B’, suffices for our purposes.

There are several groups that employ operating regimes similar to A/B/C, but this also helps keep track for readers not intimately familiar with the different variants of superconducting quantum processors. Also it might also help to summarize the $\delta\nu$ and T associated with these cases. The discussions section does some version of this, but it would be helpful to define these regimes at the start.

We have reorganized the introduction of these models in the introduction of the paper to ease an early

identification of relevant parameter regimes.

As pointed by the authors, the trade-off between increasing the qubit-qubit coupling parameter T and errors from local two-body terms (ZZ) has been long studied in superconducting qubit literature as a challenge in fixed-coupling architectures. However, this discussion needs to be much more nuanced in light of recent advances from the IBM group addressing this tradeoff (arXiv:2106.00675, arXiv:2011.07050), and its implication on the limits to increasing T .

The revised paper now provides a much substantiated picture of many facets of IBM efforts. It begins with a newly included Fig. 1 showing statistics of the parameter values for 11 of the latest IBM processors, we have reorganized our discussion of LASIQ generated structures, discuss heavily engineered frequency patterns beyond ABAB, and finish with a new paragraph (in the Discussion) commenting on the works mentioned by the referee and their efforts to tame ZZ coupling effects. (Actually, the more recent of the two preprints (arXiv:2106.00675) appeared after our preprint, and in fact *cites* our manuscript.)

A specific remark on our new descriptions of IBM work: we have chosen not to include a discussion of the small LASIQ adjustments that have been made to the qubit frequencies in Fig. 1 in the main text (but provide an extensive discussion in the Supplementary information). The elimination of certain collisions, mostly for the purpose of improving cross-resonance gate performance, does not change the MBL characteristics of the transmon array. We do give a perspective on the anticipated, more ambitious “pattern engineered” LASIQ adjustments later in the paper.

In the discussion of Fig 5: “Although definitely at a lower level, the higher-order terms of the τ - Hamiltonian are still present, with the ZZ danger threshold (dashed line) still reached. This occurs only at around $T = 9$ MHz, about three times larger than for the natural-disorder parameters. But dangerous MBL phenomenology is still lurking in the wings, even in this much more favorable setting.” Why is this ZZ danger threshold (somewhat arbitrarily defined here at 100kHz) relevant for the case of tunable couplers?

The ZZ danger threshold of 100 kHz is not arbitrary but has been established in experiments of the IBM group (see, e.g., Ku *et al.*, PRL 2020). In the new submission, we have added many new comments about ZZ coupling. We agree that tunable couplers help to reduce concerns in their regard. However, dynamic ZZ effects remain. There are also no reasons to hope that tunability reduces ZZZ coupling and next-neighbor ZZ. We therefore believe that τ -Hamiltonian framework should be considered for tunable-coupler architectures as well.

As the authors also point out, the A-B-A frequency arrangement is an oversimplification but serves to highlight some interesting features in their analysis in the limit of very small disorder. However, I don't follow this conclusion by the authors: “What is the applied significance of these observations? One is that one should not aim to reduce the frequency variance much farther.” Why is the reduced variance a problem if you want to use a more precise but more complex frequency arrangement of qubits?

More complex arrangements are possible in principle, but increasingly difficult to control in practice. (The possibility to synthetically generate localized wave functions by engineered variations of local potentials has been discussed by the MBL community, but for all we know dismissed.)

We also note that our the principal result – the existence of dangerously quantum chaotic regions for too low disorder – remains valid for more complex frequency arrangements. These include, e.g., the three-frequency A-B-A-C pattern designed to avoid the seven most likely types of ‘frequency collisions’ for transmons with CR gates on the heavy hexagon lattice (Hertzberg *et al.* npj quantum information 2021) and Chamberland *et al.* PRX 2020). This particular pattern is shown in Fig. R4(a) (and now also discussed in the manuscript). For different parts of that lattice – highlighted in Fig. R4(a) by the colored background – we calculated the IPR as a function of the disorder in E_J . As for the simplified

Figure R4 – Optimal three frequency pattern for cross-resonance architectures on the heavy-hexagon lattice. (a) 65 qubit Hummingbird processor. The colored parts are further investigated in (b), where the IPR is shown as a function of δE_J . There is no qualitative difference from the A-B pattern discussed in the manuscript (exemplified here with the results for a chain 9 geometry, gray line). Results are averaged over all states of a manifold of fixed excitation number ($N_{\text{ex}} = 6$ for geometry IV, otherwise $N_{\text{ex}} = 5$). Parameters are taken from Hertzberg *et al.* (npj quantum information 2021): $\nu_1 = 5$ GHz and $\nu_2 = 5.07$ GHz for the ‘target’ qubits (blue and red), $\nu_C = 5.14$ GHz (yellow) for the ‘control’ qubits. The (negative) anharmonicity is $E_C = 330$ MHz.

A-B pattern, there are degenerate next nearest neighbor (NNN) qubits – the ‘control’ qubits in CR language – sitting on the Kagome lattice sites (yellow). There is no qualitative difference between the new IPR results that are shown in Fig. R4(b) and the A-B results discussed in the manuscript. We observe the same sequence of disorder regions, from clean, but delocalized, states for super-weak disorder, to delocalization within multiplets and inter-multiplet hybridization, and finally to global MBL at large disorder. Depending on the exact number of degenerate NNN contained in the specific geometries, the decrease in the IPR at small δE_J is of different magnitude. For comparison, the IPR of a chain 9 geometry with an A-B pattern is also shown (dotted gray curve). The data shown here is averaged over all states of a fixed excitation manifold ($N_{\text{ex}} = 6$ for geometry IV, otherwise $N_{\text{ex}} = 5$). The IPR decreases more sharply if the analysis is restricted to only the computational multiplets, indicating that computational states are more susceptible to hybridization (in agreement with similar observations for the geometry discussed in the manuscript). The ‘inter multiplet hybridization’ (minimum for large δE_J) starts at smaller δE_J than for the chain 9 A-B configuration, because the average distance in energy between different multiplets is much smaller, due to the particular choice of the mean frequencies and the more complex multiplet structure.

In summary, the chaotic phase is still present (and even occurs at similar disorder values) even if the connectivity of the heavy hexagon lattice is lower and the frequency pattern more complex than in our manuscript. We also note that conclusions drawn from consideration of small sized clusters of two to a few qubits have limited significance for the longer ranged structure of (computational) states. Whereas CR gate performance favors reduced frequency spread (ideally down to zero), the MBL perspective requires disorder high enough to prevent their delocalization.

The authors consider 3 different indicators of chaotic fluctuations in MBL theory, but discuss that the onset of chaotic behavior is seen differently in their parameter sweeps. Can the authors comment on whether there is any understanding of why these differences arise or why certain diagnostics are not as sensitive? In the future, which diagnostic would be the go-to for such an analysis?

Our discussion of the increasing sensitivity of level statistics, inverse participation ratio and Walsh diagnostics has been further sharpened in the main text. In short, level statistics is a relatively broad-brush tool as it considers a broad range of Fock states. The Walsh diagnostics, on the other hand, focuses more selectively on the computational states and is therefore the most sensitive measure. In terms of sensitivity, it is an interesting observation that the rather high IPR threshold of about 0.9 provides very similar bounds as the Walsh diagnostic, while being considerably simpler to calculate.

Asked which of the measures would be the go-to measure, we would say that the Walsh diagnostic and the IPR diagnostic give the most sensitive information, with Walsh showing conclusively that IPR values above, say, 0.9 are safe. The Walsh results confirm that the Fock-basis IPR is indeed informative about the safety of quantum computation.

We also learned that in relatively clean devices subject to synthetic patterning, the analysis must start with an identification of physically protected subspaces distinguished, e.g., by qubit permutation symmetries. This is important because spectral statistics applied to the full Hilbert space may falsely indicate Poisson statistics (different subspaces remain uncorrelated even if energetically degenerate), even though individual states are chaotically extended. This principle, too, is now discussed in the paper.

The analysis presented in this work is undeniably unique and an interesting take on the problem, but I am struggling to find a novel insight in superconducting quantum processor design that has not been arrived at already in the traditional approach,

We tried to address this general concern in the opening paragraphs of this letter.

In addition, we would like to note that the latest IBM work [arXiv:2106.00675] has already pointed to our new perspective and its relevance for device scalability.

and find some of the conclusions questionable. For instance, in the language of the superconducting quantum computing community, increasing coupling T typically leads to large static ZZ errors, hence, one should either operate at weak T fixed couplers or make tunable T couplers.

Generally speaking, we do not mean to insinuate that tension between the principal need for effective randomness on the one hand and device efficiency on the other hand poses insurmountable difficulties. On the contrary, we hope that the diagnostic tools introduced in this paper will turn out to be useful in identifying optimized device configurations with as little disorder as possible and as much as necessary.

Concerning tunable couplers, we understand that they have a chance to fundamentally improve matters, but they are not necessarily a universal panacea, they are technologically difficult to implement, and not all groups have rushed to adopt them.

Along these lines, how is it meaningful to make a comparison of the different regimes across the same extent of T , and then conclude that the IBM strategy is more sensitive to these chaotic fluctuations? Furthermore, many of the design considerations in practice are far more nuanced compared to the discussions in this work – frequency arrangement, qubit lattice and coupling vs coherence. While some of these are acknowledged by the authors in the main body, they seem to be ignored in the rather strong concluding remarks.

At this point, we indeed do believe that the IBM approach is more vulnerable to the compromising of computational states than others, since it eschews tuning of T , and has small frequency spreads due to the needs of the CR gate. We stand by this point of view, which we tried to substantiate by a quantitative numerical analysis of various representative and present-day IBM structures in the paper. We think that it is not a coincidence that the IBM teams are putting so much work into amelioration schemes, and are coming up with ingenious work-arounds.

Metaphorically, these are like engineering a synthetic solid on the level of its unit cells. However, all experience from localization theory (and our paper) indicates that local redesign generally has consequences on larger scales which the local analysis cannot easily predict.

On the one hand, there are the principal lessons from MBL theory which we believe do extend to the many body system “transmon device”. Seen from that perspective, a transmon array is a system of interacting lattice bosons. In the absence of randomness, locally injected quantum information would immediately disperse into a continuum of extended many body wave functions. Much of our paper is about demonstrating that such compromising of information protection remains a concrete danger, including for architectures of contemporary design. On the other hand, we cannot exclude

(and hope) that there will be an ingenious engineering solution which suppresses random fluctuations, including in scaled and effectively two-dimensional layouts. In the new submission we emphasize more than before that the two views “nuanced device engineering” and “fundamental physics of many body localization” do not contradict but complement each other. For example, we have demonstrated that precision LASIQ engineering generates resonant structures beyond the nearest neighbor ZZ couplings routinely considered in transmon device physics. Techniques developed in the field of many body localization allow one to detect these resonances and to predict windows of disorder strength for optimal qubit protection. We are positive that these types of diagnostics and proposed solutions will remain applicable to future device generations, and this is the spirit in which the “rather strong” concluding remarks (now down-toned) were written.

Minor comments:

1. Figure 5b, label for contour missing (I guess it is 1/2?)

We thank the referee for spotting this and have added the label.

2. *Discussion on frequency spreads switches between δE_J and $\delta\nu$ – I would suggest sticking to one, for consistency when comparing regimes.*

We appreciate this suggestion by the reviewer and are now consistently referring to the spread of Josephson energies as δE_J in the manuscript to stay in line with the experimental parameters presented in the newly added Fig. 1.

While we aim to keep alternating references to δE_J and $\delta\nu$ at a minimum, these two define independent scales in the characterization of the transmon array (when put in relation to E_C). The occasional distinction between δE_J and $\delta\nu$ remains a necessity.

Only when turning to the discussion of LASIQ pattern engineering – whose details have now been moved to the supplementary material – we adopt our language to “frequency engineering” as it is done in the relevant publications and therefore also employ $\delta\nu$ to discuss its spread (but only after a careful introduction of this adopted language).

REVIEWERS' COMMENTS

Reviewer #1 (Remarks to the Author):

Authors of the manuscript “Transmon platform for quantum computing challenged by chaotic fluctuations” have revised and answered very thoroughly all the questions and criticism raised by us and other referees. We agree with the authors that the main criticism raised by us related on the quantum dynamics is perhaps a topic of further research. The presentation and readability have improved considerably by the structural changes and the text clarifications. We recommend the publication in Nature Communications.

Minor corrections/typos

* On the page 1: “Connections between MBL and superconducting qubits have been considered earlier [5, 6], but mainly with a focus on applications of qubit arrays as quantum simulators of the bosonic MBL transition [6].” We would like to point out that Ref 6 considers chains of semiconducting spin qubits as quantum simulators of spin MBL and Ref 5 considers chains of superconducting transmon qubits as quantum simulators of bosonic MBL. Could you please clarify/correct this in the manuscript text too?

* Figure 2: Panels (b)-(d) have reference energy indicated in top-left corner: (for example, -23.345 GHZ). Shouldn't this energy be positive, since all energies in the main panel (a) are too positive?

* Unclear comment “Define what means in Eqs (1) and (2)” was our typo/mistake in writing the first referee report, sorry about it. We meant to say “Define what $\langle i, j \rangle$ means in Eqs (1) and (2)”

* Related to “Question: Are spectral and wave function statistics and Walsh transformation results computed only for “the computational states”, i.e. the states that are highlighted with red in Fig 1(c)-(d)?”: Thank you for answering this in detail and discussing its implications thoroughly. We are happy with the discussion. However, in the current manuscript, you use two wordings either “5-excitation band” or “ $N/2=5$ bit flips” when referring to the states that are used for the spectral and wave function statistics. In our opinion, “5-excitation band” is more descriptive and accurate wording for the situation where you include all states not just the computational ones. The phrase “bit flips” intuitively refers just to local 0/1 occupations.

* For clarity, it would be good to give pointer to the definition of the Kullback-Leibler divergence (in the Methods) also in the main text. Now it just appears in the caption of Fig. 3.

* Discussion paragraph: line 513, typo “couping” → “coupling”?

Reviewer #3 (Remarks to the Author):

I appreciate the authors attempts to improve the readability of the manuscript and their incorporation of the several suggestions made by the referees. However, I still find the opening of the Diagnostics section hard to read – but I shall leave this to the discretion of the authors/editors.

I disagree with the authors on the “established threshold” of 100 kHz for the ZZ interaction and perhaps I'm missing this discussion in Ku 2020 as well. Ultimately the effect of the ZZ error has to be taken in terms of gate/circuit/idling times as well, so there isn't quite “a one size fits all” statement that can be made here. For the flux tunable couplers – can the authors refer to any discussions, if available in the literature, of NNN ZZ or next-nearest neighbor interactions? I think these are natural questions that will arise to the careful reader of Fig 4, so my suggestion to the authors would be to push some of this discussion to the main text.

To highlight their point – “the existence of dangerously quantum chaotic regions

for too low disorder”, the discussions of Fig R4 rely on extensions of ABAB.. to ABAC type frequency arrangements. However, in the limit of low disorder, even in the “more complex” arrangement, the next-nearest A -A degeneracy can be described in the language of “frequency collisions”, in the absence of an MBL inspired analysis. The authors argue that their analysis can reveal non-local interactions beyond the more “traditional” approaches in superconducting processor design to suppress ZZ. However, here too, I would expect these to be less significant if the coupling T's were reduced, (aided by improvements in coherence to maintain gate fidelity) – this was a point raised in the previous report as well (“many of the design considerations in practice are far more nuanced compared to the discussions in this workCoupling vs coherence”). The implications of such a strategy for type A devices are not discussed in the manuscript nor the rebuttal. I will however state that I do see value in the novelty of the method developed here, and am sympathetic to the authors comment on enabling the broader community to build off these ideas for future device design.

Furthermore, I assume this sentence on the apparent IBM roadmap needs to be edited on line 57, given that the authors themselves argue that such a lattice arrangement has not been employed in

IBM devices? “For instance, in its roadmap for future devices, IBM aims to replace random variations of qubit frequencies by a precision engineered frequency alternation, e.g., . . . -A-B-A-B- ..While this pattern efficiently blocks resonances between neighboring qubits, next nearest neighbors are now approximately degenerate.”

Response to the referee reports

We thank the two referees of our manuscript for their detailed and informed feedback on our revised manuscript. We appreciate the substantial amount of effort required to again work through the paper and provide detailed reports. Below we provide our response to the final points raised by Referees 1 and 3.

We have seen very high interest in our work from the community. This has again reminded us that it is a significant effort in discussions to convey all the messages of our paper when talking to people from different backgrounds (e.g. the quantum computing colleagues versus MBL experts). Reflecting this, we feel that our manuscript is already at the limit of nuance and complexity that is advisable for this first paper in what likely will be an expanding line of research. Therefore, we have adopted referee suggestions to further finesse the paper in various ways only in a limited fashion. In light of the above, we have kept our changes to the manuscript quite minimal. All editorial requests (article structure, figure features, fonts, etc.) have been taken into account, see the detailed answers in the completed “Author Checklist”.

We hope that with these amendments the revised manuscript and accompanying files are compliant with the requirements for Nature Communications.

Yours sincerely,
Christoph Berke (for all authors)

Reviewer #1 (Remarks to the Author):

Authors of the manuscript “Transmon platform for quantum computing challenged by chaotic fluctuations” have revised and answered very thoroughly all the questions and criticism raised by us and other referees. We agree with the authors that the main criticism raised by us related on the quantum dynamics is perhaps a topic of further research. The presentation and readability have improved considerably by the structural changes and the text clarifications. We recommend the publication in Nature Communications.

We thank the referee for their positive remarks and the endorsement for publication in Nature Communications.

Minor corrections/typos

- (1) *On the page 1: “Connections between MBL and superconducting qubits have been considered earlier [5, 6], but mainly with a focus on applications of qubit arrays as quantum simulators of the bosonic MBL transition [6].” We would like to point out that Ref 6 considers chains of semiconducting spin qubits as quantum simulators of spin MBL and Ref 5 considers chains of superconducting transmon qubits as quantum simulators of bosonic MBL. Could you please clarify/correct this in the manuscript text too?*

We thanks the referee for pointing us to this slight overreach of citing Ref. [6] here, which indeed deals with semiconducting spin qubits (and not transmon qubits as Ref. [5] does). As such, we decided to drop the reference to Ref. [6].

- (2) *Figure 2: Panels (b)-(d) have reference energy indicated in top-left corner: (for example, -23.345 GHZ). Shouldn't this energy be positive, since all energies in the main panel (a) are too positive?*

The referee is correct. The sign is wrong and the energies should be positive. Thanks for pointing this out. The typos are corrected in the updated version of Figure 2.

- (3) *Unclear comment “Define what means in Eqs (1) and (2)” was our typo/mistake in writing the first referee report, sorry about it. We meant to say “Define what means in Eqs (1) and (2)”*

Unfortunately, we are still left to wonder what precisely was meant by the reviewer here and are therefore unable to enact any changes to the manuscript.

- (4) *Related to “Question: Are spectral and wave function statistics and Walsh transformation results computed only for “the computational states”, i.e. the states that are highlighted with red in Fig 1(c)-(d)?”: Thank you for answering this in detail and discussing its implications thoroughly. We are happy with the discussion. However, in the current manuscript, you use two wordings either “5-excitation band” or “ $N/2=5$ bit flips” when referring to the states that are used for the spectral and wave function statistics. In our opinion, “5-excitation band” is more descriptive and accurate wording for the situation where you include all states not just the computational ones. The phrase “bit flips” intuitively refers just to local 0/1 occupations.*

We agree with the referee that the term ‘bit flips’ misleads to think only of occupations 0/1. For greater clarity, we reworded the relevant passage: ‘... states, which are generated by a total of $N/2 = 5$ bit flips and can be viewed as ...’ is replaced by ‘... states, which are generated by a total of $N/2 = 5$ excitations. [...] States within this bundle that have local excitation numbers 0,1 can be viewed as ...’

- (1) *For clarity, it would be good to give pointer to the definition of the Kullback-Leibler divergence (in the Methods) also in the main text. Now it just appears in the caption of Fig. 3.*

We thank the reviewer for this suggestion and have included such a reference to the definition of the Kullback-Leibler divergence in the main text.

- (5) *Discussion paragraph: line 513, typo “couping” → “coupling”?*

Thanks for spotting the typo, we have corrected it.

Reviewer #3 (Remarks to the Author)

I appreciate the authors attempts to improve the readability of the manuscript and their incorporation of the several suggestions made by the referees. However, I still find the opening of the Diagnostics section hard to read – but I shall leave this to the discretion of the authors/editors.

I disagree with the authors on the “established threshold” of 100 kHz for the ZZ interaction and perhaps Im missing this discussion in Ku 2020 as well. Ultimately the effect of the ZZ error has to be taken in terms of gate/circuit/idling times as well, so there isn’t quite “a one size fits all” statement that can be made here. For the flux tunable couplers – can the authors refer to any discussions, if available in the literature, of NNN ZZ or next-nearest neighbor interactions? I think these are natural questions that will arise to the careful reader of Fig 4, so my suggestion to the authors would be to push some of this discussion to the main text.

To highlight their point – “the existence of dangerously quantum chaotic regions for too low disorder”, the discussions of Fig R4 rely on extensions of ABAB.. to ABAC type frequency arrangements. However, in the limit of low disorder, even in the “more complex” arrangement, the next-nearest A -A degeneracy can be described in the language of “frequency collisions”, in the absence of an MBL inspired analysis. The authors argue that their analysis can reveal non-local interactions beyond the more “traditional” approaches in superconducting processor design to suppress ZZ. However, here too, I would expect these to be less significant if the coupling T’s were reduced, (aided by improvements in coherence to maintain gate fidelity) – this was a point raised in the previous report as well (“many of the design considerations in practice are far more nuanced compared to the discussions in this work ...Coupling vs coherence”). The implications of such a strategy for type A devices are not discussed in the manuscript nor the rebuttal. I will however state that I do see value in the novelty of the method developed here, and am sympathetic to the authors comment on enabling the broader community to build off these ideas for future device design.

Furthermore, I assume this sentence on the apparent IBM roadmap needs to be edited on line 57, given that the authors themselves argue that such a lattice arrangement has not been employed in IBM devices? “For instance, in its roadmap for future devices, IBM aims to replace random variations of qubit frequencies by a precision engineered frequency alternation, e.g., . . . -A-B-A-B- ..While this pattern efficiently blocks resonances between neighboring qubits, next nearest neighbors are now approximately degenerate.”

Our thanks to the referee for his/her reading of our manuscript, and for noting our “incorporation of the several suggestions...” Concerning their specifics, we agree that ZZ levels of 100kHz should not be viewed as an established threshold (we now change our wording in the paper to “danger threshold”); it has been more of a moving target. At the time of Ku *et al.* ’20, it was clear that good practice enabled ZZ to be in the neighborhood of 100kHz (140 is the largest documented there), but that it was clearly necessary to move to a smaller number. More recent literature has indeed focussed on smaller values of ZZ. While it goes beyond what we want to say in our paper, we note the latest on this in the tunable-coupler literature, arXiv:2112.03708 (ETH Surface-17): “we calculate residual-ZZ interaction strengths between qubits lower than 8kHz”. NNN is not reported, but is not necessarily much smaller, due to the ABABA pattern used here. It is clear that this level of ZZ is viewed with concern in this tunable-coupler work, rightly so in our opinion.

On the efficacy of MBL-inspired analysis vs. simpler “frequency collision” arguments, we respectfully disagree with the referee. Note that our mindset permits us to identify no fewer than four distinct regimes in the passage from high to low disorder in patterned arrangements – there is much more to it than frequency collisions. Our judgement was that in fact there is a little too much for the general reader to absorb here, and so have given the full story only in the Supplement, for those readers who really want to dig into the implications of the MBL-inspired analysis for these cases.

Finally, given that evidently IBM intends to use pattern strategies in upcoming processors, we see no need to change our comment on the IBM roadmap.